# Intestinal Inflammation and Regeneration–Interdigitating Processes Controlled by Dietary Lipids in Inflammatory Bowel Disease

**DOI:** 10.3390/ijms25021311

**Published:** 2024-01-21

**Authors:** Soon Jae Kwon, Muhammad Sohaib Khan, Sang Geon Kim

**Affiliations:** Integrated Research Institute for Drug Development, College of Pharmacy, Dongguk University-Seoul, Goyang-si 10326, Gyeonggi-do, Republic of Korea; ksoonj1112@gmail.com (S.J.K.); muhammadsohaibkhan786@gmail.com (M.S.K.)

**Keywords:** IBD, IECs, lipid intake, stem cells, inflammation

## Abstract

Inflammatory bowel disease (IBD), including Crohn’s disease and ulcerative colitis, is a disease of chronic inflammatory conditions of the intestinal tract due to disturbance of the inflammation and immune system. Symptoms of IBD include abdominal pain, diarrhea, bleeding, reduced weight, and fatigue. In IBD, the immune system attacks the intestinal tract’s inner wall, causing chronic inflammation and tissue damage. In particular, interlukin-6 and interlukin-17 act on immune cells, including T cells and macrophages, to amplify the immune responses so that tissue damage and morphological changes occur. Of note, excessive calorie intake and obesity also affect the immune system due to inflammation caused by lipotoxicity and changes in lipids supply. Similarly, individuals with IBD have alterations in liver function after sustained high-fat diet feeding. In addition, excess dietary fat intake, along with alterations in primary and secondary bile acids in the colon, can affect the onset and progression of IBD because inflammatory cytokines contribute to insulin resistance; the factors include the release of inflammatory cytokines, oxidative stress, and changes in intestinal microflora, which may also contribute to disease progression. However, interfering with de novo fatty acid synthase by deleting the enzyme acetyl-CoA-carboxylase 1 in intestinal epithelial cells (IEC) leads to the deficiency of epithelial crypt structures and tissue regeneration, which seems to be due to Lgr5^+^ intestinal stem cell function. Thus, conflicting reports exist regarding high-fat diet effects on IBD animal models. This review will focus on the pathological basis of the link between dietary lipids intake and IBD and will cover the currently available pharmacological approaches.

## 1. Background Information

Inflammatory bowel disease (IBD) is the immune-associated inflammation of the gastrointestinal tract. IBD includes ulcerative colitis (UC) and Crohn’s disease (CD) with persistently increasing co-morbidities [1]. The exact pathological mechanism for the progression of IBD is still a debate and includes both environmental and genetic factors [1,2]. Studies have shown that polymorphic loci regulate cytokines, chemokines signaling, and antibacterial peptides, further modulating autophagic and immune cell activity by elevating the risk of ileal or colonic CD and UC [2,3]. Clinically, IBD features disclose a variety of ranges based on age groups and gender; for instance, more than 60% of women suffering from CD also report rectal bleeding. Conversely, 62% of UC cases in males exhibit reduced rectal bleeding and abdominal pain events. The results of prevalence data show that nearly 1 million people in the USA were reported with CD. Interestingly, more than 80% of reported cases of CD reveal that the patients’ distal part of the small intestine is affected [4,5]. Nevertheless, patients suffering from UC appeared on the board with inflammatory lesions, especially in the distal part of the colon [6]. The expected increase in IBD cases might be 2.5-fold in Iran, 2.3-fold in North Africa, and 1.5-fold in India by 2035 compared to 2020 [7]. The prevalence of IBD in men was reported to be higher as compared with women [8].

### 1.1. Liver to the Gut Pathway

Gut-associated lymphoid tissue (GALT) plays a role in strengthening the mucosal immune system by acting as both protective and tolerant tissues against any pathogenic response. The liver also abundantly contains innate immune cells and acts as a primary immunological organ because of its continuous exposure to the circulating antigens and endotoxins from the gut microbiota. The portal vein takes gut-derived materials to the liver and feedback of bile from the liver to the intestine, suggestive of a reciprocal association between the microbiomes and the liver. In addition, bile acids affect the gut microbiota and interact with nuclear receptors in hepatocytes and intestinal epithelial cells to modulate metabolic activities [9]. The liver-to-gut disorders are correlated with gut and liver immune system abnormalities [10] because the liver is linked to the GALT and synergizes its immune reconnaissance [10,11]. Thus, hepatic–gut disorders comprise abnormalities of the gut and liver immune systems [10].

### 1.2. Association of Metabolic Disorder and IBD

Obesity is usually associated with a variety of metabolic syndromes, such as type 2 diabetes [12], ischemic vascular disease [13], elevated serum lipids levels, and NAFLD [14]. Any pathological disruption due to chronic caloric intake results in a metabolic syndrome (i.e., diabetes [15], obesity, and fatty liver disease [15]) [16] (Table 1).

Developing countries around the world have reported increased incidences of IBD due to Westernized lifestyles with the elevated surge of obesity [17]. The findings of the studies reveal that obesity was reported more commonly in patients with Crohn’s disease in comparison with ulcerative colitis [18]. The result of the study, including 1598 children (aged between 2 and 18 years) with IBD, reveals that 23.6% were suffering from CD while 30.1% had ulcerative colitis [19]. The prospective case-control study analysis showed that overweight/obesity was primarily common in outpatients with CD, which was ~40% of the patients [20]. Briefly, visceral obesity is interlinked with stress and leads to elevated visceral fat, which eventually leads to the activation of proinflammatory markers such as interleukins and considerably stimulates M1 macrophages [21]. Visceral adipose tissue was found to be elevated in obese patients suffering from CD and showed pronounced upregulation of various inflammatory genes such as CCL2, leptin, and IL6 [22]. In addition to this, the serum levels of adiponectin, resistin, and active ghrelin were found to be significantly elevated in IBD patients [23].

Studies report that the experimental animal models, i.e., leptin-deficient ob/ob mice, can alter the microbiome [24] to develop insulin resistance and diabetes via the regulation of several molecular cascades, such as altered fatty acid metabolism in the liver and modulation of the glucagon-like peptide [25].

The findings of several clinical and experimental findings lead to the firm belief of scientists that gut microbiota is potentially linked with Crohn’s disease and ulcerative colitis. The results of a Spanish cohort study show that dysbosis was more efficiently linked to CD patients than UC [26]. A high-fat diet and sugar combination, which mimics the effects of the Western diet, results in intestinal dysbiosis with pronounced elevation of Akkermansia, Alistopes, Bacteroides, Bilophila, Enterobacteria, and Ruminococcus torques with reduced levels of Bifidobacterium, Lactobacillus, Prevotella, and Roseburi [27].

Since the liver regulates lipid metabolism by fatty acid oxidation and lipogenesis and maintains the human body’s energy under normal physiological conditions, the metabolic disorder is often accompanied by non-alcoholic fatty liver disease (NAFLD), which was recently renamed as metabolic dysfunction-associated fatty liver disease (MAFLD), accounting for ~25% of all cases worldwide [28]. NAFLD comprises clinicopathological abnormalities and leads to steatosis with or without mild inflammation (non-alcoholic fatty liver) and a neuroinflammatory variant (non-alcoholic steatohepatitis), distinguished by the presence of hepatocellular damage such as hepatocyte ballooning [29]. The pathophysiology of liver symptoms includes lipotoxicity, autophagy dysregulation, endoplasmic reticulum stress, and IR [14], which may mechanistically account for systemic metabolic dysfunction defined by MAFLD [30,31].

**Table 1 ijms-25-01311-t001:** Association between obesity and IBD.

Number of Patients and Characteristics	Results	References
153 Crohn’s disease (CD) and 229 ulcerative colitis (UC) patients, respectively	The risk of developing CD is 2.3 times higher in obese women (18 years), and there is no significant association for UC patients with obesity	[32]
CD (*n* = 138) and UC (*n* = 394)	The risk of CD was found to be increased by 1.9 times in obese non-pregnant women, but no significant association was reported for UC	[33]
CD (*n* = 75) and UC (*n* = 177)	According to the findings of the study, obesity alone is not enough to trigger the development of either CD or UC	[34]
CD (*n* = 297) and UC (*n* = 284), respectively	No statistically significant association was found between obesity and either UC or CD in the studied patient populations	[35]
377,597 men with increased BMI have an associative risk of developing CD and UC	The results of the COX regression analysis showed a positive correlation between BMI and CD in the group of 1523 patients, while an inverse correlation was observed in UC patients (*n* = 3323)	[36]
A pooled analysis of cohort studies, including CD (*n* = 563) and UC (*n* = 1047) patients	The findings suggest that there is a significant association between CD and obese patients with BMI ≥ 30 kg/m^2^, while there is no significant relation between UC and obese patients	[17]
Systemic studies comprising 14,947 IBD subjects	Notably, 13.6% of IBD patients with NAFLD were found to have liver fibrosis	[37]

The complications of NAFLD are confined to not only T2D but also IBD [14,38]. For the first time, Thomas documented the link between colon ulceration and fatty liver early in 1873 [39]. However, the frequency of NAFLD in IBD patients varies greatly, ranging from 1.5% to 40%, depending on the diagnostic criteria [40,41]. In recent days, NAFLD progression belongs to the chronic caloric intake. Based on hospitalization diagnosis, IBD patients have a high body mass index because of high-fat diet (HFD) consumption, which exhibits deleterious effects [42]. The effect of liver disease on IBD progression is still unclear because a few dilemmas still support the theory that HFD intake damages the liver and exacerbates IBD, while others believe that HFD deleteriously affects the liver only and protects the intestine from its progression to IBD [43]. This review aims to summarize the molecular basis as to how dietary lipids intake acts as a dual sword in the case of IBD.

## 2. IBD Pathophysiology

### 2.1. Intestinal Permeability and Barrier

Human health depends on the structural veracity of epithelial and endothelial barriers in the body. The intestine accompanies the largest internal barrier and takes part in body protection against the harmful chemicals and bacteria found in the gut. The barrier comprises the mucus layer, commensal bacteria, epithelial cells, and immune cells in the lamina propria [44]. Intestinal epithelial goblet cells conceal mucus glycoproteins and inhibit the micro-organism and colonocytes’ direct contact with the gut, whereas mucus released in the small intestine allows bacteria to move freely [45]. Paneth cells are responsible for releasing anti-microbial proteins, which neutralize the effects of bacterial cells in the small intestine, whereas B cells secrete IgA in the lamina propria, which binds to bacteria, and its related toxins hinder their entry into the body [46].

Recent scientific advancements have significantly improved our understanding of IEC functions and their subtypes, such as enterocytes and goblet cells [47]. However, intestinal enteroendocrine cells comprise various subgroups, including enterochromaffin cells, D cells, and G cells [47,48]. Gunnar C. Hansson and his colleagues discovered a new subtype of goblet cells and named it sentinel goblet cells found at the apex of colonic crypts [49]. Unlike traditional goblet cells, the cells have a distinct ability to sense bacteria and respond by secreting mucin, which results in the adjacent environment becoming red in response to noxious stimuli [49]. Tuft cells exhibit cellular differentiation and exist in two subtypes: one type of epithelial cell expresses cytokine Tslp, whereas the other one articulates the immunological marker CD45 [50].

Germ-free animal studies reveal that microbes play a role in fostering appropriate intestinal development and function. Germ-free mice have thin mucosa, resulting in diminished IEC proliferation and compromised fabrication of mucins and other IEC-producing mediators [51]. Because of the loss of the mucin layer, germ-free mice result in the candid disclosure of colitogenic toxins, emphasizing the gut microbiome’s function in intestinal tissue protection and healing [52]. The reduction of microbiota and their substitution by pathogens, known as dysbiosis, may have the capability to alter the gut barrier. Intestinal nutrients and water absorption occur through the transcellular and paracellular pathways and junctions. Intestinal pore channels are charged and size-selective, so pore size varies; the lowest limit is 8 Å diameter, whereas the largest diameter is ~100 Å, a non-selective leaky pathway [53,54].

### 2.2. Inflammatory Mediators

Although several variables are engaged in the pathophysiology of IBD, a disruption in the epithelial barrier is primarily found. The initial injury causes inflammation, causing additional damage and a vicious spiral. Tumor necrosis factor α (TNF-α) is a prototype proinflammatory cytokine released by activated macrophages, monocytes, and T lymphocytes. Study results on CD patients found enhanced TNF-α proteins and mRNA levels in mucosal biopsies [55].

TNF is majorly found via actuated macrophages and T lymphocytes having 26 kDa. TNF binds to TNF receptor 1 and leads to the generation of TNF receptor signaling complex (complex-I), which encompasses the core proteins, TRADD [56], TRAF2, RIPK1, cIAP1/2, and the linear ubiquitin chain assembly complex [57]. The results of the experimental studies have proven that TNF-α is involved in mucosal inflammation in CD [58]. TNF-α modulates gut inflammation in CD patients via a variety of mechanisms. In in vitro experiments using patient specimens in clinical trials of anti-TNF-α therapy, the levels of TNF-α were found to be reduced with subsequential downregulation of IFN-γ in the mucosa [59]. Hence, TNF-neutralizing monoclonal antibodies (e.g., vedolizumab) are used to deal with CD and UC [60], whereas antibodies to IL-12/IL-23 p40 are for the management of CD [61,62].

The interleukin family of cytokines (i.e., IL-1α, IL-1β, IL-18, IL-33, and IL-36) play roles in the modulation of the proinflammatory pathway, resulting in intestinal inflammation through NF-κB [63,64,65]. Studies have reported a significant drop in the IL-1 receptor antagonists to IL-1 ratio in IBD patients, suggestive of the significance of the IL-1 pathway in the exacerbation of IBD [66]. Furthermore, in severe infant-onset IBD and animal experiments, decreased IL-10 signaling, which may account for the enhanced production of IL-1 in macrophages, also results in CD4^+^ T cell activation [67,68,69]. IL-10-related cytokines, such as IL-19, IL-20, IL-22, IL-24, IL-26, IL-28, and IL-29, are all involved in the modulation of inflammatory and immune responses (Figure 1) [70]. In mice studies, IL-1β may be a potential inducer of Helicobacter hepaticus-mediated colitis by stimulating innate lymphoid cells and conscription of neutrophils via the IL-1 receptor signaling pathway [71], which modulates mucosal aggregation of T cells and produces TH17, resulting in colitis [72,73] and further carcinogenesis [74]. Retinoic acid-related orphan receptor-γt (RORγt) is a transcription factor that is involved in the development of TH17 [75]. The results from other studies reveal that IL-1β activation leads to chronic intestinal inflammation by endorsing the accretion of IL-17A-secreting innate lymphoid cells and CD4^+^ Th17 cells, resulting in intestinal pathology [76]. Consistently, IL-1 receptor antagonist treatment in mice ameliorates acute colitis [77].

IL-6 is generated by several cells present inside the tumor, such as tumor-infiltrating cells and stromal cells. IL-6 in normal blood concentration (1.6 pg/mL) facilitates a mild immune response against the defense of incessant pathogens [78]. Studies have proven that IL-6 association with the vagus nerve may have effects on the smooth muscle cells or secretory cells, which results in intestinal motility and secretion [78,79]. The classic pathway of IL-6 signaling includes the binding of IL-6 with the membrane-bound receptor IL-6 receptor-α, also known as IL-6R. This binding results in the development of a heterohexameric complex comprising two IL-6, IL-6R, and the β subunit of IL-6 receptor (gp130) [80,81]. This complex then leads to the stimulation of the JAK/STAT3 pathway, consequently integrating STAT3 target genes (Figure 2). Interestingly, the complex also triggers the PI3K/AKT/mTOR and RAS/RAF/MEK/ERK pathways [82]. The major role of the classical pathway is to provoke anti-inflammatory impacts during the acute-phase response [83].

IL-6 also induces a trans-signaling cascade, including soluble IL-R6 (sIL-R6) binding to IL-6. sIL-6 is produced due to alternative splicing of IL-6R mRNA or via the breakdown of membrane-bound IL-R6 through ADAM 10 or ADAM 17 [84,85]. The interaction of IL-6 to sIL-6R results in a complex formation, provoking the dimerization of gp130 and stimulating the downstream signaling cascade (Figure 2). The complex of IL-6/IL-6R is bound by disulfide bonds and activates Box-1 and Box-2 in the cytoplasmic domain of gp130; this results in JAK activation leading to phosphorylation at a tyrosine residue of gp130 cytoplasmic domain [86]. The phosphorylated pTyr-X-X-Gln motif (X = amino acid) on gp130 conscripts Src homology domain in STAT3. Phosphorylation of STAT3 in response to IL-6 impedes the binding of Suppressors of Cytokine Signaling 3 (SOCS3) to STAT3 (Figure 2) [87]. The phosphorylation of STAT3 by JAK at the tyrosine residue leads to STAT3 dimerization and nuclear translocation and target gene transcription (i.e., intestinal inflammation and cancer) [88,89,90]. The result of another study shows that STAT3 and STAT4 act reciprocally on intestinal inflammation. The stimulation of STAT4 via either IL-12 or leukemia inhibitory factor via STAT3 inhibits the action of Th17 and promotes the repair of intestinal epithelial damage in IBD.

Interestingly, gp130 chain dimerization by the IL-6-IL6R complex stimulates the non-overlapping intracellular signaling pathway via phosphorylation of the cytoplasmic region of gp130 linked with the Janus kinase family (Figure 2). The resultant stimulation leads to activator protein 1 (AP-1) phosphorylation and induces the inflammatory genes [91,92].

IL-9 is initially found as a synergistic growth factor for T and mast cells and plays a role in asthma [93]. Studies have shown that IL-9 mRNA levels were raised in UC patients. Mechanistically, IL-9 is secreted by peripheral blood lymphocytes and binds to its receptor in the gut and polymorphonuclear leukocytes. Astonishingly, IL-9 stimulation potentiates IL-8. IL-9 is overexpressed in epithelial cells and activates STAT5 [94].

In addition, IL-18 is normally in a proactive form and is stimulated by the action of cleavage enzyme caspase-1 into the stimulation of NLRP3, which enhances the risk of metabolic and autoimmune disorders [95,96]. NLRP3 inflammasome activation has been demonstrated in patients with IBD. Systematically, IL-18 is produced via overstimulated intestinal epithelial cells or macrophages, leading to goblet-cell depletion and churning out proinflammatory cytokines, including IFN-γ and TNF-α [97,98].

Recently, IL-36α and IL-36γ levels have been shown to be significantly increased in patients with IBD. In experimental studies, IL-36γ was identified in intestinal epithelium nuclei, while IL-36α was detected in CD14^+^ inflammatory macrophages in the cytoplasm [99]. Reduced IL-36 levels potentiated the dextran sodium sulfate (DSS)-induced acute colitis by impairing cell proliferation and enhancing the effect of IL-22 associated with fibroblast stimulation [99,100]. In another study of humans with IBD, fibrotic intestinal tissues showed enhanced levels of IL36A, which is responsible for the regulation of the genes involved in the fibrogenesis in fibroblast. However, a reciprocal effect was observed in IL-36 knockout mice treated with either 2,4,6-trinitrobenzene sulfonic acid (TNBS) or DSS, showing diminished chronic colitis and intestinal fibrosis [101].

Interleukin-37 has anti-inflammatory and innate immunity suppressor activity, while the IL-37b epithelial expression level was raised in IBD patients. Experimentally, IL-37b blocks the TNF-α-induced-interferon-γ-inducible protein-10 expression in human colonic subepithelial myofibroblasts [102]. Overexpression of IL-37 reduces DSS-induced colitis in transgenic mice [103]. Another study’s results demonstrate that the IL-37b gene transfer by an adenovirus vector causes the potentiation of mesenchymal stem cells against DSS-induced colitis via stimulating the Treg cell activity and inhibiting cytokines release [104].

### 2.3. Immune Mechanisms

Patients with IBD lack resistance to enteric commensal bacteria and show macrophage, neutrophil, and T/B cell responses [105,106]. Resistance is facilitated in normal hosts by governing T/B lymphocytes, NK cells, and dendritic cells [58,107]. TNF and IL-12 p40 have been related to the etiology of CD in antibody-neutralization studies, whereas T cells have been associated with UC by T-cell-ablative medications [108] such as cyclosporin and tacrolimus [109,110]. Recent study findings show that reduced epithelial expression of microbiota-sensitive histone deacetylase 3 (HDAC3) leads to the elevated accretion of commensal-specific CD4^+^ T cells in the intestine [111]. In both CD and UC, the cells engaged in innate responses are triggered, resulting in the enhanced production of cytokines and chemokines. In all types of IBD, macrophages and dendritic cells in the lamina propria are augmented in an absolute quantity. However, in CD, TH1- and TH17-related cytokines implicated in innate immunity are preferentially activated and have been rarely reported in UC [112] (Figure 3). Moreover, TH17-associated transcription factor RORγt levels were elevated in the lamina propria of IBD patients [75].

TLRs on the cell membrane bind to bacterial and viral targets. TLRs are least expressed in the normal physiological intestinal environment, whereas in the case of pathogenesis, TLRs are expressed in intestinal, respiratory, and urogenital epithelial cells. Overexpression of TLRs in the epithelial layer leads to the enhanced release of cytokines, chemokines, and anti-microbial peptides. Ligand activation of TLR then activates the NF-κB and MAPK signaling pathways [107,113]; the transcription factors promote pro- and anti-inflammatory gene expression. CARD4 (also known as NOD1) and CARD15 (formerly NOD2) homologous intracellular receptors bind to diaminopimelic acid and muramyl dipeptide to activate NF-κB. TLR2 activation affects CARD15 and NF-κB activation [114,115]. Most of the cytokines can be selectively inhibited to delay colitis.

The epithelial layer is the primary line of defense against infections. Epithelial chemokines can be found on the luminal surface of the vascular endothelium in both local tissue and draining lymph nodes and there, they contribute to cell recruitment [116]. Monocytes and PMNs bind to injury sites, generating more proinflammatory mediators than resident macrophages. IBD patients show increases in proinflammatory cytokines and overexpression of adhesion molecules and co-stimulatory molecules [110]. To facilitate the migration of the cells, ICAM1 is required for cells in the blood to adhere and activate endothelium. It is worth noting that adhesion molecules, such as ICAM-1, have been shown to bind CD11b/CD18. However, ICAM-1 is produced only on the apical epithelial surface during inflammation [117]. Intestinal macrophages that reside in the gut have a reduced capacity to respond to bacterial components. This leads to the dysregulation of bacterial eminent receptors, including TLRs and CD14, which act as co-ligands of LPS [118].

### 2.4. Lipids and Inflammation

The dysregulation of lymphocyte trafficking and immune cell migration can lead to the dissemination of chronic inflammation, prompting researchers to investigate potential agents responsible for lymphocyte migration and infiltration via the bloodstream to inflamed targets in the intestinal mucosa. One area of interest is sphingolipids (S1Ps), which are active metabolic products involved in the inflammatory cascade and immune response. S1Ps contribute to the maintenance of structural components of eukaryotic membranes, and their cascades participate in de novo synthesis and catabolic recycling with various physiological functions, leading to the recruitment of lymphocytes in injurious areas of the intestine, which intensifies inflammation by enhancing proinflammatory cytokines [39].

### 2.5. Intestinal Stem Cell Niche and Cell Signaling

Acute inflammation kills Lgr5^+^ stem cells in both the small intestine and the colon [119]. Infections caused by bacteria, viruses, or parasites can damage significant regions of the gut, such as a plethora of crypt–villus units [120]. In addition, radiation, chemotherapeutic drugs, and antibiotics all cause intestinal injury in the crypts and villi [121]. Since the developed cells at the villi show a short life cycle of a few days, removing the stem or its progenitor cells in the crypts impairs the recovery of epithelial cells in the wound-associated lesion [122,123]. Since the cells do not perform all of the necessary intestinal tasks, this is only a temporary solution. The cells are then replaced within a week by creating functioning crypts from scratch. The fission of freshly generated crypts is necessary to compensate for lost crypts on a massive scale [124,125], a process slowed by crypt fusion [126].

Intestinal stem cells interact physically with those having epithelial and mesenchymal nature. The intestinal stem cell niche is made up of these cells and their interactions. Damage can be generated in several ways to examine the regeneration response. High-dose radiation exhausts Lgr5^+^ cells [127], which has been experimentally used to explore the regeneration response [128]. DSS, an experimental agent used to induce both acute and chronic colitis [129], also elicits crypt loss [130]. Using an animal model, the targeted knock-in of Lgr5 has been shown to provide a sophisticated strategy for understanding stem cell biology [131]. Interestingly, Lgr5^+^ specific deletion did not bring any visual changes in intestine architecture [131,132].

Stem cells inside the niche are subdivided and compete for limited niche space [133,134]. As a result, cells towards the niche’s boundary are more likely to be pushed out of the niche [134,135]. Consequently, stem cell-promoting stimuli facilitate cell differentiation to progenitors, which move towards the transit-amplifying zone at crypt compartments. They thereafter undergo many cycles of cell division. The most common lineage option involves choosing between the secretory and absorptive lineages [136]. The results of the in vivo experiments show that the intestinal epithelium is sustained after Paneth cell depletion, which supports the alternate cascade activation [137,138]. Thus, a decrease in Paneth cells corresponds to a decrease in stem cells; their retention is required in the activity of in vitro stem cells [46]. WNT, specifically WNT3, EGF, and DLL4, are produced in epithelial Paneth cells [46,139], which assists stem cell metabolism by providing lactate as a substrate for oxidative phosphorylation [140].

Enteroendocrine and tuft cells may substitute for deleted PCs and provide a juxtaposed origin of Notch signals, even though the mesenchyme adjoining the epithelium releases adequate concentrations of Wnt ligands [141,142]. In the gut, the mesenchymal compartment comprises fibroblasts, creating extracellular matrix components and myofibroblasts [143]. Many subpopulations have been shown to assist stem cells. Gli1^+^ cells express Wnt2b, whereas CD34^+^ cells express Rspo1 and Wnt2b [144]. Foxl1^+^ cells that express Wnt2b and Rspo3 and Pdgfra^+^ myofibroblasts are examples [145,146]. Mesenchymal cells thereby contribute to the activity of intestinal stem cells by establishing a healthy gradient of BMP signaling [147]. According to McCarthy et al. (2020), Pdgfra1^low^ mesenchymal cells secrete gremlin 1. On the other hand, mesenchymal telocytes Pdgfra1^high^ are found in the villus, and after their stimulation, they promote BMP signaling [148]. Apart from Paneth cells, deep crypt cells in the colon activate Notch [149]. Because canonical Wnt ligands are not generated in the epithelium, they must be obtained from the adjacent mesenchyme. As a result, when Wnt-secreting Gli1^+^ mesenchymal cells in the colon are diminished, the colonic architecture collapses [150].

### 2.6. Role of Extracellular Matrix in Intestinal Regeneration

Healthy cells do not proliferate in fluid, implying that they require anchoring to a solid matrix [151]. Thus, the characteristics of the extracellular matrix (ECM) have a substantial impact on the cells. Cells can assess ECM stiffness via receptors such as integrins and adjust their intracellular significance as per requirements. The stiffness is a critical parameter for stem cell differentiation [152]. Consequently, the ECM regulates cellular characteristics such as differentiation.

YAP/TAZ is considered the primary effector of stiffness sensing, contributing to intestinal regeneration and stem cell proliferation. In an in vitro model, matrix stiffening leads to the activation of YAP [153]. Under the umbrella of matrix stiffness, YAP/TAZ modulates the transition to proliferation in mammary epithelial cells [154]. The results of in vitro experiments reveal that intestinal stem cells demand a rigid matrix for optimal growth [155]. Gα12/13-coupled receptors block the Hippo pathway kinases (i.e., Lats1/2), triggering YAP and TAZ transcription coactivators [156].

It has been shown that DSS treatment in the genetic model with deletion of integrin leads to a protective effect [157]. FAK-YAP-mTOR signaling cascade may be involved in the proliferation and differentiation of progenitor cell tissue [158].

## 3. Dietary Lipids and IBD Progression

### 3.1. The Effects of Fats on IBD

The incidence of IBD is increasing concurrently with the rise in overweight and obesity rates. Contrary to traditional belief, a notable portion of IBD patients (31.5%) are obese, and this could potentially be linked to the development and progression of IBD [159]. Cystic fibrosis, another example of a disease characterized by extensive intestinal damage due to a genetic deficiency in CFTR, decreases fluid production in epithelial cells [160]. CF patients with increased inflammation and damage at the static mucus layer lead to ineffective protection against bacterial infections [161].

A recent published study including seven case-control and two prospective cohorts comprising 1491 IBD patients and 5309 normal subjects reveals that the Western diet is associated with the progression of IBD [162]. Experimentally, animal models fed with high-fat diets have been systematically studied, and scientific evidence shows that consumption of a Western diet, which is high in fat, is directly linked to increased inflammation in the large intestine. Recent experimental studies have shown that mice fed a Western diet were protected from colonic inflammation compared to those fed a normal diet [43]. However, whether fat consumption triggers defensive cascades against DSS-induced inflammatory pathways or impedes the well-known DSS-induced colitis remains unclear.

It should be noted that the metabolic functions of fatty acids exhibit distinct characteristics in the function of intestinal epithelial cells. Although there is uncertainty surrounding the cellular activity of fatty acid oxidation, fatty acid synthesis has been shown to have an impact on intestinal epithelial cells. Studies suggest that acetyl-CoA-carboxylase 1-mediated FAS contributes to the maintenance of Lgr5^+^ stem cell function. As a result, FAS promotes the production of organoids and the differentiation of crypt structures by maintaining PPARδ/β-catenin [163]. Inhibition of the FAS pathway in intestinal epithelial cells reduced epithelial crypt structures and decreased Lgr5^+^ intestinal epithelial stem cells (Figure 4).

On the contrary, excess fat intake disturbs the phospholipid membrane structure of the epithelial cells. Intestines absorb lipids from the intestinal mucosa. In fasting conditions, the small intestine efficiently uses plasma fatty acid for oxidation and esterification and increases the uptake capacity of triacylglycerol absorption up to six-fold [164]. Triacylglycerol hydrolysis by pancreatic lipase then yields 2-monoacyglycerol (2-MAG), which is engulfed by the intestinal enterocytes [165], whereas esterified cholesterol hydrolyzed by means of cholesterol esterase generates cholesterol and fatty acid. The resulting cholesterol is taken up into micelles, which mainly contain bile acids, along with lower levels of phospholipids, FFAs, and 2-MAG [165,166]. The micelles are absorbed into enterocytes via the brush border, where they secrete fatty acid and 2-MAG and are absorbed, where it takes part in synthesizing chylomicrons. However, dietary and biliary lipids produce lysophosphatidylcholine and free fatty acids under the action of pancreatic phospholipase A2 [167].

### 3.2. The Effects of Bile Acids on IBD

Primary bile acids (PBA) production occurs in the liver via two cascades (i.e., classical and alternative pathways) (Figure 5). Cholic acid (CA) and chenodeoxycholic acid (CDCA) are prevalently abundant PBAs in humans. The classical pathway produces approximately 90% of the bile acid [168]. The 7α-hydroxyl group interacts with cholesterol to produce 7α-hydroxycholesterol in the presence of cytochrome P450s (CYPs). The production of 7α-hydroxycholesterol is a rate-limiting step catalyzed by CYP7A1 [169]. The CYP7A1 gene ciphers an enzyme named cholesterol 7α-hydroxylase, which is responsible for cholesterol breakdown and bile acid synthesis [170]. Consistently, the results from animal studies show that homozygous deletion mutation in CYP7A1 resulted in hyperlipidemia [170]. It has also been shown that bile acid production is elevated in DSS-induced IBD due to a compensatory increase in CYP7A1 [171].

BA is transported into the small intestine via the ampulla of Vater in the second portion of the small intestine and accelerates the reabsorption of lipid molecules in the jejunum [172]. As BA is unable to get absorbed by the small intestine, it leads to the release of a significant portion (more than 90%) of BA through the small intestine (Figure 5), which is resorbed by the hepatic portal vein and named enterohepatic circulation of BAs [172]. The remnants of bile acids within the intestine undergo a series of chemical transformations, including deconjugation, desulfation, dehydrogenation, dehydroxylation, and isomerization, facilitated by colonic bacteria [173,174].

PBAs dehydoxylation at carbon-7 leads to deconjugation and produces secondary bile acids (SBAs) such as lithocholic acid (LCA), deoxycholic acid (DCA), and ursodeoxycholic acid [175]. The recent mechanistic flow of converting PBAs, i.e., CA and CDCA into SBAs, LCA, and DCA, may be explained by the group of 7α-dehydroxylating bile-acid-induced (bai) operon enzymes naming BaiB, BaiCD, BaiA2, BaiE, BaiF, and BaiH found within Clostridium cluster XIVa species including Lachnospiraceae and Ruminococcaceae families, and also Eubacterium species [176].

In general, seven species are involved in bacterial/microbial clade, while Subdoligranulum, Gemmiger, and Faecalibacterium genera hold close links as they are involved in butyrate production, which is found to have beneficial effects on IBD. However, clade production is decreased by the physiological and immunological reactions, consequently aggravating the IBD [177]. A group of Subdoligranulum species is involved in forming new clades and has been found to reduce IBD and IBD-linked metabolites such as bile acids and polyunsaturated fatty acids [178]. Summarizing the strain-level reporting of interlinked micro-organisms with host epithelium reveals the organ-specific microbial species accountable for the IBD-allied surge of primary unconjugated bile acids and diminution of SBAs [177]. Post-cholecystectomy patients’ fecal BAs and mucosal microbiome analysis showed elevated immuno-regulatory activity and SBA negatively associated with peripheral monocyte levels [179]. The study’s results, including 14 healthy control patients and 39 CD patients, show that there were significantly low levels of SBA, LCA, and DCA observed in the serum and fecal of CD patients. Moreover, Enterobacteriaceae and Lachnospiraceae were robustly found in patients with CD, resulting in psychological comorbidity by disturbing their bile acids metabolism [180].

Conjugated bile acids make micelles with lipids containing phosphatidylcholine and cholesterol, while in the stomach, a gastric enzyme lipase acts on the dietary lipids and converts them into diacylglycerol and fatty acids [181]. Gastric lipase is different from pancreatic lipase but has a close resemblance. Newborn infants have low levels of pancreatic lipase, which puts emphasis on the alternative mechanism for fat digestion fulfilled by extra-pancreatic lipases (i.e., gastric lipase and ligual lipase in humans and rats) to meet the physiological demands [182,183]. The fatty acids in the stomach assist in the emulsification of lipids, followed by their movement towards the small intestine, where they are further emulsified via bile acids, strengthening the lipolytic activity of pancreatic lipases. Alternatively, the small intestine promotes the production of microbes that either promote lipid absorption or inhibit lipid intake [183,184]. These events may facilitate epithelial cell turnover, if in excess, injuries, potentially aggravating IBD progression. Hence, it remains to be established what the exact roles of appropriate amounts of dietary fat supplies and types of lipids are for the prevention of injury and promotion of stem cell regeneration.

## 4. Therapeutic Approaches and New Candidates

The available treatments for IBD, including filgotinib, tofacitinib, infliximab, and adalimumab, along with others, are being practiced by physicians to relief the symptoms of IBD (Table 2).

In addition, new drug candidates for IBD are described below:The standard therapy for IBD patients typically involves the use of aminosalicylates [195] and corticosteroids [196], which have been utilized for many years to alleviate pain and inflammation.It has been shown that IL-6 was raised in mice treated with 5% DSS-induced acute colitis, while IL-6 was reduced after treatment with SM934 (artemisinin analog) and ameliorated experimental colitis [197]. The result of the tocilizumab trial on 36 patients with CD has been reported with clinical significance [198]In experimental models, it has been observed that deficiency of monoacylglycerol acyltransferase (MGAT) 2 provides protection against obesity. Moreover, the specific deletion of MGAT2 deters fat accumulation in the intestine [199]. In another study, monoacylglycerol lipase (MAGL) inhibition enhances the 2-arachidonoglycerol levels and results in decreased macroscopic and histological colon alterations, lowering cytokine levels [200]. MGAT2 deficiency in the intestine safeguards mice from metabolic disorders induced by high-fat feeding [201]. JTP-103237, currently in the preclinical stage, is an inhibitor of MGAT2 and impairs the absorption of luminal lipids in mice [202]. TNBS-induced murine colitis was reversed by the potent MAGL inhibitor JZL184 [200], and another MAGL inhibitor URB602 significantly repressed whole gut transient [203].Recent research suggests that ketogenic diets (KD) can increase the levels of circulating ketone bodies and have an anti-inflammatory effect [204]. However, the effects of this particular diet on colitis are still not well-understood. Animal studies have been conducted using KD, a low-carbohydrate diet, and a normal diet [204]. Following colitis, KD was found to protect intestinal barrier function and reduce inflammatory cytokines. Thus, KD may alleviate colitis by modifying microbiota.IBD frequently leads to liver injury. Milk fat globule membrane (MFGM) has been shown to mitigate colitis and liver injury [205]. Prophylactic MFGM therapy was found to be effective against colitis, improving weight loss, disease activity index, and pathological scores. Moreover, MFGM reduced levels of inflammatory mediators with an increase in IL-10 levels. MFGM thus alleviated DSS-induced injury, enhancing the mucosal barrier. It appears that MFGM may decrease oxidative stress in the liver [205].Signaling agents, including Wnt, EGF, Notch, and BMP ligands, promote the proliferation of Lgr5^+^ stem cells [206].Sphingosine-1-phosphate (S1P) is a signaling molecule involved in physiological processes. In IBD, the excessive infiltration of immune cells into the intestinal tissue is a significant contributor to the pathogenesis of the disease. Studies have shown that targeting S1P receptors could be a viable therapeutic strategy for IBD. Monoclonal antibodies directed to S1P have been tested in preclinical models of prostate and kidney cancer, but no studies have been conducted in IBD [207,208,209]. However, S1P receptor modulators have shown promising effects in preclinical studies [210] and are currently being evaluated in clinical trials for inflammatory disorders. These agents work downstream of S1P receptors to limit lymphocyte recruitment to inflammatory areas, reducing immune cell infiltration and mitigating inflammation in the intestine. Recently, it was reported that ozanimod has been in phase II for CD and phase III for UC treatment. Etrasimod is currently in phase II trials for UC, while amiselimod has completed phase II trials for CD [211].Therapeutic agents that enhance insulin sensitivity, such as GLP-1, SGLT-2, and PPAR-γ ligands, have shown benefits for IBD patients by improving insulin-sensitized supplies of fuel and building block sources [212]. However, the potential impact of obesity on IBD treatment efficacy is still not well understood. Studies on various autoimmune diseases suggest that obesity can significantly affect therapeutic efficacy, leading to suboptimal treatment outcomes due to rapid clearance and decreased trough concentrations of medications. Therefore, further investigation is needed to better understand the interplay between obesity and IBD treatment outcomes.Aryl hydrocarbon receptor (AhR) activation upregulates IL-22 production, which may protect the intestine from inflammation [213]. Vegetables like broccoli and cabbage can stimulate AhR, which is highly expressed in intestinal intraepithelial lymphocytes and may be involved in the protection against luminal attacks [214].Formula-defined feed enteral nutrition showed positive results in CD patients, with 40% relapse chances within 6 months [215].

## 5. Conclusions

IBD results from the dysregulated immune system and the release of inflammatory mediators and lipotoxicity. Since inflammatory cytokines and lipotoxicity contribute to insulin resistance generation, the patients with IBD and those with metabolic disorders have common characteristics in the context of proinflammatory cytokines and oxidative stress.Inhibition of de novo FAS affects intestinal stem cell function and regeneration capacity, so the intake of dietary lipids should be carefully interpreted to understand epithelial tissue repair and regeneration for IBD patients (Figure 6).Anti-inflammatory agents and insulin-sensitizing drugs are therapeutically beneficial to patients with IBD due to the inhibition of inflammatory injury, efficient cellular fuel oxidation, and increased tissue regeneration capacity.

## Figures and Tables

**Figure 1 ijms-25-01311-f001:**
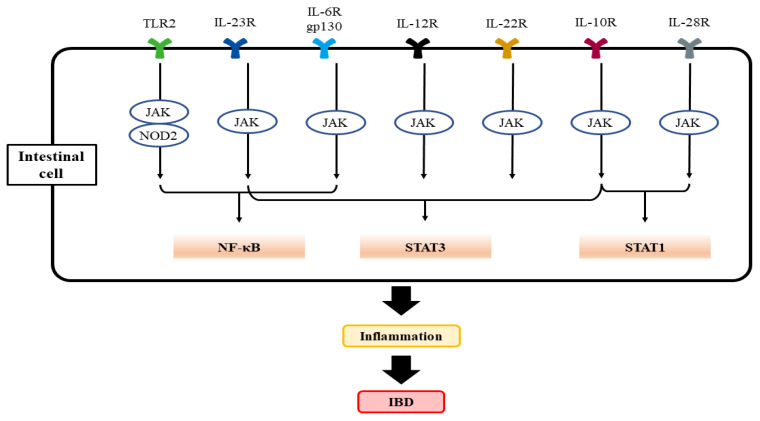
A scheme showing TLR2 and interleukin-mediated activation of key transcription factors responsible for the activation of genes associated with IBD progression. CARD15 (formerly NOD2) and homologous intracellular receptors bind to diaminopimelic acid and muramyl dipeptide to activate NF-κB. Abbreviations: TLR2, toll-like receptor 2; IL-6R, interlukin-6 receptor; CARD15, caspase activating recruitment domain 15; NF-κB, nuclear factor kappa-light-chain-enhancer of activated B cells.

**Figure 2 ijms-25-01311-f002:**
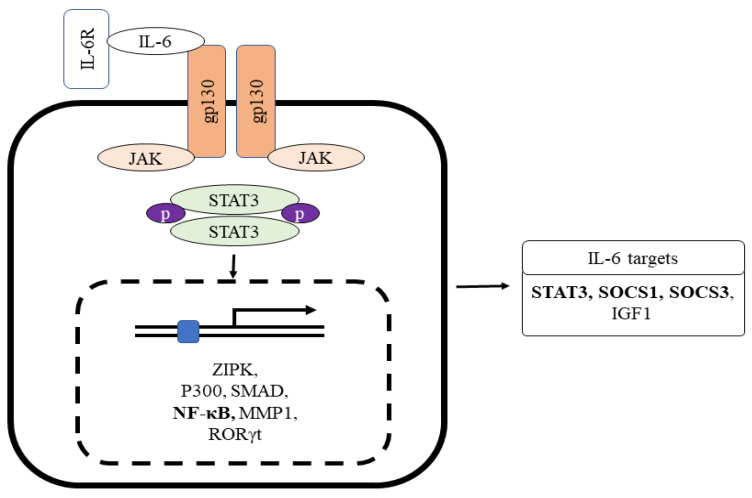
The role of IL-6R interaction with gp130 for Stat3-mediated inflammatory target gene expression. Gp130 chain dimerization by the IL-6-IL6R complex stimulates the non-overlapping intracellular signaling pathway via phosphorylation of the cytoplasmic region of gp130 linked with the Janus kinase family. Abbreviations: IL-6, interlukin-6; IL-6R, interlukin-6 receptor; gp130, glycoprotein 130.

**Figure 3 ijms-25-01311-f003:**
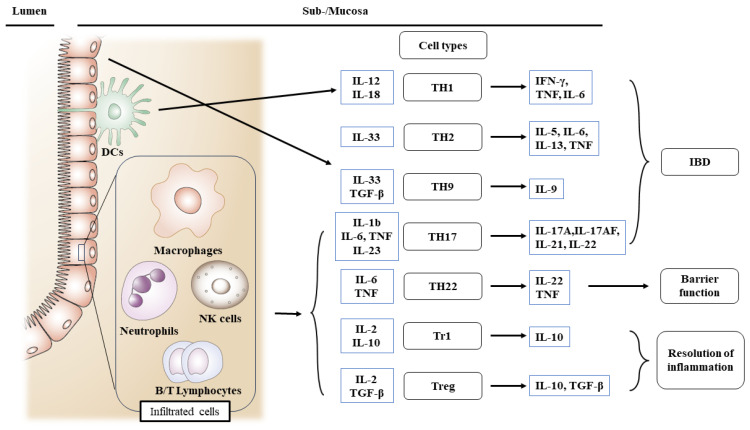
Roles of infiltrating inflammatory cells for the activation of different T cell subsets engaged in intestinal injury, barrier, and inflammation resolution in IBD. IBD increases the production of inflammatory mediators. In IBD, cells engaged in innate immune responses are triggered. In all types of IBD, macrophages and dendritic cells are augmented in an absolute quantity. Abbreviations: IBD, inflammatory bowel disease; TH, T helper cells; DC, dendritic cells; NK, natural killer cells; Treg, regulatory T cells.

**Figure 4 ijms-25-01311-f004:**
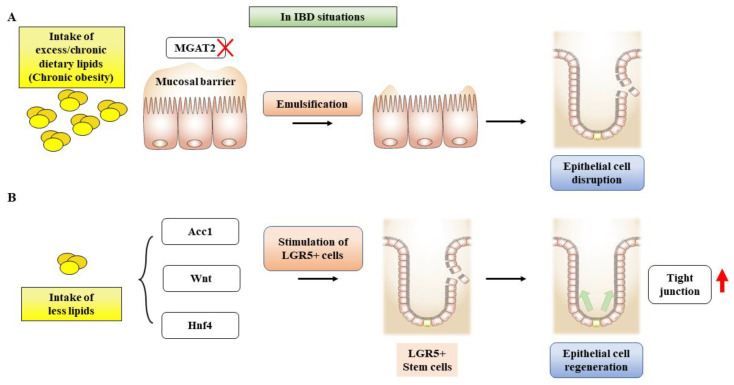
A scheme showing the effects of dietary lipids on intestinal stem cell growth and differentiation. (**A**) MGAT2 deficiency inhibits fat accumulation in the intestine, protecting IECs in an animal model with diet-induced obesity and glucose intolerance. (**B**) Inhibition of de novo FAS in IECs results in a deficiency of epithelial crypt structures, which in turn leads to a reduction in Lgr5^+^ stem cells (ISCs). Abbreviations: MGAT2, monoacylglycerol acyltransferase 2; IECs, intestinal epithelial cells; FAS, fatty acid synthase, LGR5^+^, leucine-rich repeat containing G protein-coupled receptor 5.

**Figure 5 ijms-25-01311-f005:**
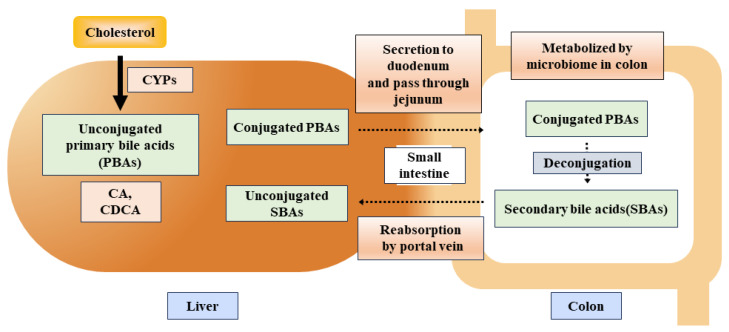
A schematic flow of primary and secondary bile acids in gastrointestinal processes and microbial interactions. Abbreviations: PBAs, primary bile acids; SBAs, secondary bile acids; CA, cholic acid; CDCA, chenodeoxycholic acid.

**Figure 6 ijms-25-01311-f006:**
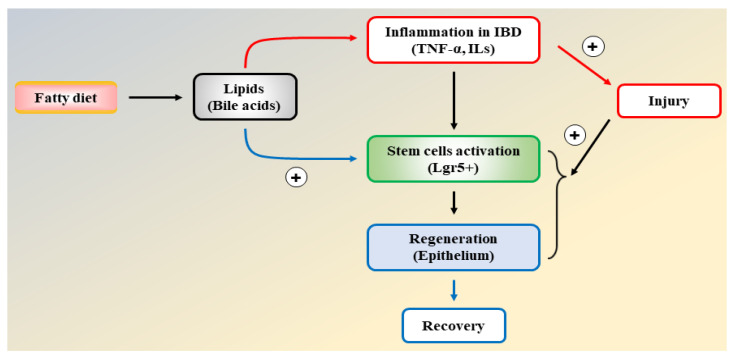
A flow chart illustrating the interdigitating effects of lipids on inflammation and intestinal epithelial regeneration.

**Table 2 ijms-25-01311-t002:** Drugs available for IBD treatment.

Drugs	Mechanisms of Action	Doses	References
Adalimumab	Monoclonal antibody to TNF-α	Subcutaneous injection 5–10 µg/mL leads to reduced TNF-α levels	[185]
Filgotinib	JAK1 inhibitor	100 mg O.D. While 200 mg resulted in a primary embolism	[186,187]
Golimumab	Monoclonal antibody to TNF-α	Initial starting with 200 mg and reduced to 100 mg after 2 weeks, and the dose is maintained by either 50 mg or 100 mg administered at intervals of 4 weeks for UC treatment	[188]
Infliximab	Monoclonal antibody to TNF-α	Highest blood concentration via intravenous infusion 80–100 µg/mL and not less than 5 µg/in 4–6 weeks	[185]
Mesalazine (5-ASA)	Anti-inflammatory effect on colonic epithelial cells	0.5 g and can be increased to 1 g 5-aminosalicylic acid T.I.D. against ulcerative colitis	[189]
Methotrexate	Inhibition of the enzymes responsible for nucleotide synthesis	12.5–25 mg/week p.o or i.p.	[190]
Tofacitinib	JAK1, JAK3 inhibitor	5 or 10 mg B.I.D. for moderately to severe UC	[191,192,193]
Vedolizumab	Anti-α4β7 integrin	300 mg within 2 weeks	[194]

## Data Availability

This manuscript has no associated data.

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
