# Peer review of "Intestinal Inflammation and Regeneration–Interdigitating Processes Controlled by Dietary Lipids in Inflammatory Bowel Disease"

_ijms, 2024, doi:10.3390/ijms25021311_

Round 1

Reviewer 1 Report (New Reviewer)

Comments and Suggestions for Authors

I recommend the publication of manuscript ijms-2795115 “Interdigitating Mechanism of Lipids Involved in Inflammation and Intestinal Regeneration” in IJMS after a major revision.

In my opinion, English language is understandable, and the work does not require any language editing. Authors review very interesting topic about the impact of nutrition and lipids on inflammation and pathophysiology in IBD.

Below I present the most important remarks regarding the work.

MAJOR ISSUES:

1.     The title does not reflect what is described in the text. Since the authors devote a large part of their work to IBD (chapter 2) and the impact of dietary lipids on IBD (chapter 3), I suggest changing and specifying the title, e.g “Intestinal Inflammation and Regeneration - two processes controlled by dietary lipids in Inflammatory Bowel Diseases (IBD)”, or “Influence of dietary lipids on inflammation and regeneration in Inflammatory Bowel Diseases (IBD)” or other, more specific one.

2.     Although the work is extensive and contains a large amount of references, it seems chaotic and does not always contain appropriate references to a given text. I recommend reading and organizing the work, and adding appropriate references where necessary.

3.     Some aspects that should be described in a given chapter are either omitted or poorly described, e.g.:

- fatty acids are essential components of dietary lipids. Classification divides them into short-chain fatty acids (SCFAs), medium-chain fatty acids (MCFAs), long-chain fatty acids (LCFAs), and very long-chain fatty acids (VLCFAs). LCFAs and VLCFAs are mainly derived from dietary intake, whereas SCFAs are formed by the conversion of indigestible dietary fibers by specific gut bacteria. The manuscript contains superficial information on this topic, even though it is extensively described by other authors in their publication.

- authors write in line 61: “Any pathological disruption due to chronic caloric intake results in metabolic syndrome (i.e., diabetes, obesity, and fatty liver disease)”. Due to the topic of the work, please describe more detail about the relationship between obesity and IBD, as well as the impact of diet and microbiota composition, in patients with IBD. Some information appears in this work but is scattered in different subsections, please describe it in one place/ paragraph/ maybe subchapter?

- add information about the effect of the Western diet on IBD, not only the work of Papoutsis et al. [30].

4. Add more information about “Dietary interventions and their therapeutic role in IBD”.

MINOR ISSUES:

1.     Please provide in the Introduction statistics on IBD (not only the incidence in the USA) as well as its prevalence in women vs men.

2.     My suggestion of subtle changes to the Figures:

-        improve the quality of Figure 2 and 5 – there are illegible, please also correct the format and style to match the rest in the manuscript.

-        in Figure 1 – mark on the diagram in the membranes of which cells these receptors are anchored. The reader may have trouble reading whether these are intestinal cells, macrophages or else? This is also not explained in the text.

-        in Figure 3   what about the involvement of neutrophils and ILC1? which cytokines do they secrete? Also label the epithelial layer cells, and add DC and NK cells, TH and Treg cells, into the abbreviations under the Figure.

-        in Figure 6 - expand with examples, e.g. which diet, which stem cells, which lipids?

3.     If according to the authors the title was supposed to be “Interdigitating Mechanism of Lipids Involved in Inflammation and Intestinal Regeneration”, then subsection 2.4 “Lipids and inflammation” should be one of the most important, and there are a few lines and only one reference [26]…

4.     I propose to introduce subchapters in Chapter 3. “Dietary Lipids and IBD progression” and describe examples there, this will make the work transparent and more readable.

5.     In subchapter 2.2. “Inflammatory mediators” the authors describe TNFα, IL-6, IL-9, IL-18, IL-36 and IL-37, but what about the rest cytokines from Fig 1? Shouldn't this description refer to the Figure? Why are these particular ones described in the text and not others? Please explain or organize the text.

6.     In line 61: “Any pathological disruption due to chronic caloric intake results in metabolic syndrome (i.e., diabetes, obesity, and fatty liver disease) [9]” – at the end, the authors cited the work Principi et al [9], however, it only applies to nonalcoholic fatty liver disease. Please apply the references separately to each metabolic syndrome, e.g. diabetes [X], obesity [Y], and fatty liver disease [Z].

7.     In line 77: “Thus, NAFLD, recently renamed as metabolic dysfunction-associated fatty liver disease (MAFLD)” add appropriate references, e.g:

a)     Gill MG, Majumdar A. Metabolic associated fatty liver disease: Addressing a new era in liver transplantation. World J Hepatol. 2020 Dec 27;12(12):1168-1181. doi: 10.4254/wjh.v12.i12.1168.      or/ and

b)     Alharthi J, Latchoumanin O, George J, Eslam M. Macrophages in metabolic associated fatty liver disease. World J Gastroenterol. 2020 Apr 28;26(16):1861-1878. doi: 10.3748/wjg.v26.i16.1861.

8.     In line 143: it should be IL-1β and not IL-β.

9.     Add a references in chapter 2.5 “Intestinal stem cell niche and cell signaling”:

a)     Karmakar S, Deng L, He XC, Li L. Intestinal epithelial regeneration: active versus reserve stem cells and plasticity mechanisms. Am J Physiol Gastrointest Liver Physiol. 2020 Apr 1;318(4):G796-G802. doi: 10.1152/ajpgi.00126.2019.

This recommendations only refines the overall draft of the manuscript.

The manuscript submitted by the Authors is in line with the subject of the IJMS, and will be an attractive article for the Readers.

Author Response

Response to Referees Letter

Reviewer 1 Comments

I recommend the publication of manuscript ijms-2795115 "Interdigitating Mechanism of Lipids Involved in Inflammation and Intestinal Regeneration" in IJMS after a major revision. In my opinion, English language is understandable, and the work does not require any language editing. Authors review very interesting topic about the impact of nutrition and lipids on inflammation and pathophysiology in IBD. Below I present the most important remarks regarding the work.

MAJOR ISSUES:

  1. The title does not reflect what is described in the text. Since the authors devote a large part of their work to IBD (chapter 2) and the impact of dietary lipids on IBD (chapter 3), I suggest changing and specifying the title, e.g "Intestinal Inflammation and Regeneration - two processes controlled by dietary lipids in Inflammatory Bowel Diseases (IBD)", or "Influence of dietary lipids on inflammation and regeneration in Inflammatory Bowel Diseases (IBD)" or other, more specific one.

        Answer: The authors would like to thank the reviewer for valuable feedback. We have changed the article title to "Intestinal Inflammation and Regeneration-Interdigitating Processes Controlled by Dietary Lipids in Inflammatory Bowel Disease."

  1. Although the work is extensive and contains a large amount of references, it seems chaotic and does not always contain appropriate references to a given text. I recommend reading and organizing the work, and adding appropriate references where necessary.

        Answer: The authors would like to thank the reviewer for valuable comment. The references are counter-checked and placed appropriately in their place to maintain the transparency of the authors' work recognition.

  1. Some aspects that should be described in a given chapter are either omitted or poorly described, e.g.: - fatty acids are essential components of dietary lipids. Classification divides them into short-chain fatty acids (SCFAs), medium-chain fatty acids (MCFAs), long-chain fatty acids (LCFAs), and very long-chain fatty acids (VLCFAs). LCFAs and VLCFAs are mainly derived from dietary intake, whereas SCFAs are formed by the conversion of indigestible dietary fibers by specific gut bacteria. The manuscript contains superficial information on this topic, even though it is extensively described by other authors in their publication.

        Answer: The authors tried best to eliminate the above points in the revised MS.

  1. Authors write in line 61: "Any pathological disruption due to chronic caloric intake results in metabolic syndrome (i.e., diabetes, obesity, and fatty liver disease)". Due to the topic of the work, please describe more detail about the relationship between obesity and IBD, as well as the impact of diet and microbiota composition, in patients with IBD. Some information appears in this work but is scattered in different subsections, please describe it in one place/ paragraph/ maybe subchapter?

        Answer: The authors would like to appreciate the valuable insight of the reviewer. We added the required details in the manuscript. The edition includes the following.

        (1) Developing countries around the world have reported increased incidences of IBD due to Westernized lifestyles with the elevated surge of obesity [1]. The findings of the studies reveal that obesity was reported more commonly in patients with Crohn's disease in comparison with ulcerative colitis [2]. The result of the study, including 1598 children (aged between 2-18 years) with IBD, reveals that 23.6 % were suffering from CD while 30.1% had ulcerative colitis [3]. The prospective case-control study analysis showed that overweight/obesity was primarily common in outpatients with CD, which was approximately 40% of the patients [4]. Briefly, visceral obesity is interlinked with stress and leads to elevated visceral fat in the body, which eventually leads to the activation of pro-inflammatory markers such as interleukins and considerably stimulates M1 macrophages [5]. Visceral adipose tissue was found to be elevated in obese patients suffering from CD and showed pronounced upregulation of various inflammatory genes such as CCL2, leptin, and IL6 [6]. In addition to this, the serum levels of adiponectin, resistin, and active ghrelin were found to be significantly elevated in IBD patients [7].

        (2) The findings of several clinical and experimental findings lead to the firm belief of scientists that gut microbiota is potentially linked with Crohn's disease and ulcerative colitis. The results of a Spanish cohort study show that dysbosis was more efficiently linked to CD patients than UC [8]. High-fat diet and sugar combination, which mimics the effects of the Western Diet, results in intestinal dysbiosis with pronounced elevation of Akkermansia, Alistopes, Bacteroides, Bilophila, Enterobacteria, and Ruminococcus torques with reduced levels of Bifidobacterium, Lactobacillus, Prevotella and Roseburi [9].

  1. Add information about the effect of the Western diet on IBD, not only the work of Papoutsis et al. [30]. Answer: The authors acknowledge the reviewer's effort to improve the quality of the manuscript. We added the required portion for the manuscript and highlighted it for your convenience.

        High-fat diet and sugar combination, which mimics the effects of the Western Diet, results in intestinal dysbiosis with pronounced elevation of Akkermansia, Alistopes, Bacteroides, Bilophila, Enterobacteria, and Ruminococcus torques with reduced levels of Bifidobacterium, Lactobacillus, Prevotella and Roseburi [9]. A recent study published including seven case-control and two prospective cohorts comprising 1491 IBD patients and 5309 normal subjects reveals that the Western diet is associated with the progression of IBD [10].

  1. Add more information about "Dietary interventions and their therapeutic role in IBD".

        Answer: The authors acknowledge the reviewer's effort to improve the quality of the manuscript. We added the dietary intervention and their therapeutic role in the manuscript.

        Aryl hydrocarbon receptor (AhR) activation upregulates IL-22 production, which further protects the intestine from inflammation [11]. Vegetables like broccoli and cabbage can stimulate AhR, which is highly expressed in intestinal intraepithelial lymphocytes and is involved in the protection against luminal attacks [12].  Formula-defined feed enteral nutrition showed positive results in CD patients, with 40% relapse chances within 6 months [13].

MINOR ISSUES:

Please provide in the Introduction statistics on IBD (not only the incidence in the USA) as well as its prevalence in women vs men.

        Answer:    We added the required portion: The expected increase in IBD cases will be 2.5-fold in Iran, 2.3-fold in North Africa, and 1.5-fold in India by 2035 compared to 2020 [14]. The prevalence of IBD in men was reported to be higher as compared with women [15].

My suggestion of subtle changes to the Figures: Improve the quality of Figure 2 and 5 – there are illegible, please also correct the format and style to match the rest in the manuscript.

        Answer: We have improved Figures 2 and 5 in the edited manuscript.

In Figure 1 – mark on the diagram in the membranes of which cells these receptors are anchored. The reader may have trouble reading whether these are intestinal cells, macrophages or else? This is also not explained in the text.

        Answer: We added the required information on the cells in the edited Figure 1.

In Figure 3  –  what about the involvement of neutrophils and ILC1? which cytokines do they secrete? Also label the epithelial layer cells, and add DC and NK cells, TH and Treg cells, into the abbreviations under the Figure.

        Answer: Abbreviations were added in the edited Figure 3. TH, T helper cells; DC, Dendritic cells; NK, natural killer cells; Treg, Regulatory T cells

In Figure 6 - expand with examples, e.g. which diet, which stem cells, which lipids?

        Answer: We added the specific diet and stem cells.

If according to the authors, the title was supposed to be "Interdigitating Mechanism of Lipids Involved in Inflammation and Intestinal Regeneration", then subsection 2.4 "Lipids and inflammation" should be one of the most important, and there are a few lines and only one reference [26]…

        Answer: The authors would like to thank the reviewer for the suggestions. We have improved the title, and because of that, we find it unnecessary to give a new subsection.

I propose to introduce subchapters in Chapter 3. "Dietary Lipids and IBD progression" and describe examples there, this will make the work transparent and more readable.

        Answer: According to the comment, the authors included subtitles in the edited version.

In subchapter 2.2. "Inflammatory mediators" the authors describe TNFα, IL-6, IL-9, IL-18, IL-36 and IL-37, but what about the rest cytokines from Fig 1? Shouldn't this description refer to the Figure? Why are these particular ones described in the text and not others? Please explain or organize the text.

        Answer: The authors would like to thank the reviewer for the suggestions. We updated the Figure and removed the unnecessary portions.

In line 61: "Any pathological disruption due to chronic caloric intake results in metabolic syndrome (i.e., diabetes, obesity, and fatty liver disease) [9]" – at the end, the authors cited the work Principi et al [9], however, it only applies to nonalcoholic fatty liver disease. Please apply the references separately to each metabolic syndrome, e.g. diabetes [X], obesity [Y], and fatty liver disease [Z].

        Answer: The authors thank the reviewer's comments for improving the quality of the manuscript. We added the references. Any pathological disruption due to chronic caloric intake results in metabolic syndrome (i.e., diabetes [16], obesity, and fatty liver disease [16]) [17].

In line 77: "Thus, NAFLD, recently renamed as metabolic dysfunction-associated fatty liver disease (MAFLD)" add appropriate references, e.g: a) Gill MG, Majumdar A. Metabolic associated fatty liver disease: Addressing a new era in liver transplantation. World J Hepatol. 2020 Dec 27;12(12):1168-1181. doi: 10.4254/wjh.v12.i12.1168. 

        Answer: The authors thank the reviewer's comment. We added the reference given as per your suggestion. Thus, NAFLD, recently renamed as metabolic dysfunction-associated fatty liver disease (MAFLD), may cause systemic metabolic dysfunction [18, 19].

In line 143: it should be IL-1β and not IL-β.

        Answer: We corrected typos in the manuscript.

Add a references in chapter 2.5 "Intestinal stem cell niche and cell signaling": a)   Karmakar S, Deng L, He XC, Li L. Intestinal epithelial regeneration: active versus reserve stem cells and plasticity mechanisms. Am J Physiol Gastrointest Liver Physiol. 2020 Apr 1;318(4):G796-G802. doi: 10.1152/ajpgi.00126.2019.

        Answer: The authors thank the reviewer's comments for improving the quality of the manuscript. We added the reference given as per your suggestion.

        WNT, specifically WNT3, EGF, and DLL4, are produced in epithelial Paneth cells [20, 21], which assists stem cell metabolism by providing lactate as a substrate for oxidative phosphorylation [22].

        This recommendations only refines the overall draft of the manuscript. The manuscript submitted by the Authors is in line with the subject of the IJMS, and will be an attractive article for the Readers.

References

  1. Chan, S. S. M.; Chen, Y.; Casey, K.; Olen, O.; Ludvigsson, J. F.; Carbonnel, F.; Oldenburg, B.; Gunter, M. J.; Tjønneland, A.; et al. Obesity is associated with increased risk of Crohn’s disease, but not ulcerative colitis: a pooled analysis of five prospective cohort studies. Clin Gastroenterol Hepatol 2022, 20, (5), 1048-1058. https://doi.org/10.1016/j.cgh.2021.06.049
  2. Mendall, M. A.; Viran Gunasekera, A.; Joseph John, B.; Kumar, D. Is obesity a risk factor for Crohn’s disease? Dig Dis Sci 2011, 56, 837-844. https://doi.org/10.1007/s10620-010-1541-6
  3. Long, M. D.; Crandall, W. V.; Leibowitz, I. H.; Duffy, L.; Del Rosario, F.; Kim, S. C.; Integlia, M. J.; Berman, J.; Grunow, J.; et al. Prevalence and epidemiology of overweight and obesity in children with inflammatory bowel disease. Inflamm Bowel Dis 2011, 17, (10), 2162-2168. https://doi.org/10.1002/ibd.21585
  4. Nic Suibhne, T.; Raftery, T. C.; McMahon, O.; Walsh, C.; O'Morain, C.; O'Sullivan, M. High prevalence of overweight and obesity in adults with Crohn's disease: associations with disease and lifestyle factors. J Crohns Colitis 2013, 7, (7), e241-e248. https://doi.org/10.1016/j.crohns.2012.09.009
  5. Balistreri, C. R.; Caruso, C.; Candore, G. The role of adipose tissue and adipokines in obesity-related inflammatory diseases. Mediators Inflamm 2010, 2010. https://doi.org/10.1155/2010/802078
  6. Zulian, A.; Cancello, R.; Micheletto, G.; Gentilini, D.; Gilardini, L.; Danelli, P.; Invitti, C. Visceral adipocytes: old actors in obesity and new protagonists in Crohn's disease? Gut 2011, gutjnl-2011. https://doi.org/10.1136/gutjnl-2011-301354
  7. Karmiris, K.; Koutroubakis, I. E.; Xidakis, C.; Polychronaki, M.; Voudouri, T.; Kouroumalis, E. A. Circulating levels of leptin, adiponectin, resistin, and ghrelin in inflammatory bowel disease. Inflamm Bowel Dis 2006, 12, (2), 100-105. https://doi.org/10.1097/01.MIB.0000200345.38837.46
  8. Pascal, V.; Pozuelo, M.; Borruel, N.; Casellas, F.; Campos, D.; Santiago, A.; Martinez, X.; Varela, E.; Sarrabayrouse, G.; et al. A microbial signature for Crohn's disease. Gut 2017, gutjnl-2016. http://dx.doi.org/10.1136/gutjnl-2016-313235
  9. Godala, M.; Gaszyńska, E.; Zatorski, H.; Małecka-Wojciesko, E. Dietary interventions in inflammatory bowel disease. Nutrients 2022, 14, (20), 4261. https://doi.org/10.3390/nu14204261
  10. Zhao, M.; Feng, R.; Ben‐Horin, S.; Zhuang, X.; Tian, Z.; Li, X.; Ma, R.; Mao, R.; Qiu, Y.; et al. Systematic review with meta‐analysis: environmental and dietary differences of inflammatory bowel disease in Eastern and Western populations. Aliment Pharmacol Ther 2022, 55, (3), 266-276. https://doi.org/10.1111/1751-2980.12910
  11. Monteleone, I.; Rizzo, A.; Sarra, M.; Sica, G.; Sileri, P.; Biancone, L.; MacDonald, T. T.; Pallone, F.; Monteleone, G. Aryl hydrocarbon receptor-induced signals up-regulate IL-22 production and inhibit inflammation in the gastrointestinal tract. Gastroenterology 2011, 141, (1), 237-248. https://doi.org/10.1053/j.gastro.2011.04.007
  12. Li, Y.; Innocentin, S.; Withers, D. R.; Roberts, N. A.; Gallagher, A. R.; Grigorieva, E. F.; Wilhelm, C.; Veldhoen, M. Exogenous stimuli maintain intraepithelial lymphocytes via aryl hydrocarbon receptor activation. Cell 2011, 147, (3), 629-640. https://doi.org/10.1016/j.cell.2011.09.025
  13. Richman, E.; Rhodes, J. M. Evidence‐based dietary advice for patients with inflammatory bowel disease. Aliment Pharmacol Ther 2013, 38, (10), 1156-1171. https://doi.org/10.1111/apt.12500
  14. Olfatifar, M.; Zali, M. R.; Pourhoseingholi, M. A.; Balaii, H.; Ghavami, S. B.; Ivanchuk, M.; Ivanchuk, P.; Nazari, S. H.; Shahrokh, S.; et al. The emerging epidemic of inflammatory bowel disease in Asia and Iran by 2035: A modeling study. BMC Gastroenterol 2021, 21, (1), 204. https://doi.org/10.1186/s12876-021-01745-1
  15. Barberio, B.; Massimi, D.; Cazzagon, N.; Zingone, F.; Ford, A. C.; Savarino, E. V. Prevalence of primary sclerosing cholangitis in patients with inflammatory bowel disease: a systematic review and meta-analysis. Gastroenterology 2021, 161, (6), 1865-1877. https://doi.org/10.1053/j.gastro.2021.08.032
  16. Lo, L.; McLennan, S. V.; Williams, P. F.; Bonner, J.; Chowdhury, S.; McCaughan, G. W.; Gorrell, M. D.; Yue, D. K.; Twigg, S. M. Diabetes is a progression factor for hepatic fibrosis in a high fat fed mouse obesity model of non-alcoholic steatohepatitis. J Hepatol 2011, 55, (2), 435-444. https://doi.org/10.1016/j.jhep.2010.10.039
  17. Principi, M.; Iannone, A.; Losurdo, G.; Mangia, M.; Shahini, E.; Albano, F.; Rizzi, S. F.; La Fortezza, R. F.; Lovero, R.; et al. Nonalcoholic fatty liver disease in inflammatory bowel disease: prevalence and risk factors. Inflamm Bowel Dis 2018, 24, (7), 1589-1596. https://doi.org/10.1093/ibd/izy051
  18. Ekstedt, M.; Hagström, H.; Nasr, P.; Fredrikson, M.; Stål, P.; Kechagias, S.; Hultcrantz, R. Fibrosis stage is the strongest predictor for disease‐specific mortality in NAFLD after up to 33 years of follow‐up. Hepatology 2015, 61, (5), 1547-1554. https://doi.org/10.1002/hep.27368
  19. Gill, M. G.; Majumdar, A. Metabolic associated fatty liver disease: Addressing a new era in liver transplantation. World J Hepatol 2020, 12, (12), 1168. http://dx.doi.org/10.4254/wjh.v12.i12.1168
  20. Sato, T.; Van Es, J. H.; Snippert, H. J.; Stange, D. E.; Vries, R. G.; Van Den Born, M.; Barker, N.; Shroyer, N. F.; Van De Wetering, M.; et al. Paneth cells constitute the niche for Lgr5 stem cells in intestinal crypts. Nature 2011, 469, (7330), 415-418. https://doi.org/10.1038/nature09637
  21. Karmakar, S.; Deng, L.; He, X. C.; Li, L. The Engineered Gut: Use of Stem Cells and Tissue Engineering to Study Physiological Mechanisms and Disease Processes: Intestinal epithelial regeneration: active versus reserve stem cells and plasticity mechanisms. Am J Physiol Gastrointest Liver Physiol 2020, 318, (4), G796. https://doi.org/10.1152/ajpgi.00126.2019
  22. Rodríguez-Colman, M. J.; Schewe, M.; Meerlo, M.; Stigter, E.; Gerrits, J.; Pras-Raves, M.; Sacchetti, A.; Hornsveld, M.; Oost, K. C.; et al. Interplay between metabolic identities in the intestinal crypt supports stem cell function. Nature 2017, 543, (7645), 424-427. https://doi.org/10.1038/nature21673

Dear Editor,

On behalf of the authors, I would like to express sincere thanks for your positive decision on our manuscript considered in the special issue of “Intestinal Inflammation and Regeneration-Interdigitating Processes Controlled by Dietary Lipids in Inflammatory Bowel Disease.” We put more efforts into revising the MS according to the helpful comments of the referees. I appreciate your kind final consideration for disseminating our paper in the International Journal of Molecular Sciences

Sincerely,

Professor Sang Geon Kim, Ph.D.

College of Pharmacy

Dongguk University-Seoul

Reviewer 2 Report (New Reviewer)

Comments and Suggestions for Authors

Somehow, the title does not mention IBD as the review's primary focus; I would change the current title to reflect that.

I would also emphasize that the description of the function of lipids in various physiologic processes, including inflammation and tissue regeneration, is not new, albeit somewhat controversial.

Fig. 1 shows activation primarily. It would be nice for the readers if the authors would clarify if these pathways are, in fact, promoting inflammation. 

In Fig. 2, the font size should be bigger because it is hard to read in its current form.

Fig 3. has the same problem when it comes to readability. Furthermore, the balance of the listed actions is the key when it comes to developing IBD and moving forward with the disease toward resolution. From this figure, these balancing actions are not really clear.

Regarding dietary lipids and IBD progression, it should be noted that characterizing the Western diet with its high-fat content only is not entirely correct.

Is the presentation and impression of Fig. 4., actually supported by previous papers regarding the beneficial effects of MGAT2 deficiency or inhibition on the development of IBD? Would this also be true with moderate lipid consumption? This is not clear from Fig. 4.

Fig. 5 is OK.

Table 2.: it would be nice to include the year when the listed drug was first used among patients. Moreover, It should be listed with every drug if it is working better against Crohn's disease or ulcerative colitis, or both.

Fig. 6 is a very important one. However, it is not clear how the two pathways from lipids would lead to injury or recovery, more specifics would be helpful.

Comments on the Quality of English Language

English needs minor editing only.

Author Response

Response to Referees Letter

Reviewer 2 Comments

  1. Somehow, the title does not mention IBD as the review's primary focus; I would change the current title to reflect that. I would also emphasize that the description of the function of lipids in various physiologic processes, including inflammation and tissue regeneration, is not new, albeit somewhat controversial. Fig. 1 shows activation primarily. It would be nice for the readers if the authors would clarify if these pathways are, in fact, promoting inflammation. 

       Answer: The authors would like to thank the reviewer for the helpful comments. We added the information about the cells in the revised Figure 1.

  1. In Fig. 2, the font size should be bigger because it is hard to read in its current form.

        Answer: Font size was changed in the revised Figure 2.

  1. Fig 3. has the same problem when it comes to readability. Furthermore, the balance of the listed actions is the key when it comes to developing IBD and moving forward with the disease toward resolution. From this Figure, these balancing actions are not really clear.

        Answer: We also edited Figure 3 in the manuscript.

  1. Regarding dietary lipids and IBD progression, it should be noted that characterizing the Western diet with its high-fat content only is not entirely correct.

        Answer: The authors acknowledge the reviewer's effort to improve the quality of the manuscript. We added the required portion for the manuscript and highlighted it for your convenience.

        High-fat diet and sugar combination, which mimics the effects of the Western Diet, results in intestinal dysbiosis with pronounced elevation of Akkermansia, Alistopes, Bacteroides, Bilophila, Enterobacteria, and Ruminococcus torques with reduced levels of Bifidobacterium, Lactobacillus, Prevotella and Roseburi [1]. A recent study published including seven case-control and two prospective cohorts comprising 1491 IBD patients and 5309 normal subjects reveals that the Western diet is associated with the progression of IBD [2].

  1. Is the presentation and impression of Fig. 4., actually supported by previous papers regarding the beneficial effects of MGAT2 deficiency or inhibition on the development of IBD? Would this also be true with moderate lipid consumption? This is not clear from Fig. 4.

        Answer: The authors would like to thank the reviewer for the suggestions. MGAT2 presence is not linked with the IBD progression. We revised Figure 4 appropriately.

  1. Table 2.: it would be nice to include the year when the listed drug was first used among patients. Moreover, It should be listed with every drug if it is working better against Crohn's disease or ulcerative colitis, or both.

        Answer: Table 2 shows the symptomatic medicine classification. As the biomarkers for inflammation like TNFa are found to be increased in CD and UC, cumulatively called IBD, we used the general term for them and mentioned CD or UC wherever necessary.

  1. Fig. 6 is a very important one. However, it is not clear how the two pathways from lipids would lead to injury or recovery, more specifics would be helpful.

        Answer: The authors would like to thank the reviewer for his/her suggestions. We improved and edited figure 6 by the addition of specific details.

References

  1. Godala, M.; Gaszyńska, E.; Zatorski, H.; Małecka-Wojciesko, E. Dietary interventions in inflammatory bowel disease. Nutrients 2022, 14, (20), 4261. https://doi.org/10.3390/nu14204261
  2. Zhao, M.; Feng, R.; Ben‐Horin, S.; Zhuang, X.; Tian, Z.; Li, X.; Ma, R.; Mao, R.; Qiu, Y.; et al. Systematic review with meta‐analysis: environmental and dietary differences of inflammatory bowel disease in Eastern and Western populations. Aliment Pharmacol Ther 2022, 55, (3), 266-276. https://doi.org/10.1111/1751-2980.12910

Dear Editor,

On behalf of the authors, I would like to express sincere thanks for your positive decision on our manuscript considered in the special issue of “Intestinal Inflammation and Regeneration-Interdigitating Processes Controlled by Dietary Lipids in Inflammatory Bowel Disease.” We put more efforts into revising the MS according to the helpful comments of the referees. I appreciate your kind final consideration for disseminating our paper in the International Journal of Molecular Sciences

Sincerely,

Professor Sang Geon Kim, Ph.D.

College of Pharmacy

Dongguk University-Seoul

Round 2

Reviewer 1 Report (New Reviewer)

Comments and Suggestions for Authors

Accept in present form.

This manuscript is a resubmission of an earlier submission. The following is a list of the peer review reports and author responses from that submission.

Round 1

Reviewer 1 Report

Comments and Suggestions for Authors

-      -   “Clinically, IBD characteristics vary 36 in age groups and on a gender basis; i.e., >60% of female patients exhibit CD with more 37 rectal bleeding, whereas 62% of males have UC with comparatively less rectal bleeding 38 and abdominal pain [4]. More than 80% of the reported CD cases affect the distal part of 39 the small intestine. The prevalence data estimates that ~1 million people in the USA suffer 40 from CD [5]. However, inflammatory lesions were observed in the distal colon of patients 41 suffering from UC [6].”

These sentences are a mess and have poor adherence to real clinical picture of the two diseases.

-        * “Thus, NAFLD, known as MAFLD”

NALFD is the old term. Now the term is MAFLD

-       - Increase the size of the characters of Figure 1, 4

-        -The treatments of IBD in Table 2 are randomly ordered and randomly chosen (incomplete)

-        - A clinician co-author expert in IBD could be useful in improving the clinical part of the review

-        - “The standard therapy for IBD patients typically involves the use of aminosalicylates”

Are you talking about ulcerative colitis or Crohn’s disease?

-        - “The result of the tocilizumab trial on 36 patients 463 with CD has been reported with clinical significance [180]”

Tocilizumab failed in IBD

-        - “However, S1P receptor 495 modulators have shown promise in preclinical studies [192] and are currently being 496 evaluated in clinical trials for inflammatory disorders.”

Not updated data

Comments on the Quality of English Language

Minor correction

Author Response

Response to Referees Letter
Reviewer 1

Comments and Suggestions for Authors:

-  “Clinically, IBD characteristics vary 36 in age groups and on a gender basis; i.e., >60% of female patients exhibit CD with more 37 rectal bleeding, whereas 62% of males have UC with comparatively less rectal bleeding 38 and abdominal pain [4]. More than 80% of the reported CD cases affect the distal part of 39 the small intestine. The prevalence data estimates that ~1 million people in the USA suffer 40 from CD [5]. However, inflammatory lesions were observed in the distal colon of patients 41 suffering from UC [6].”

These sentences are a mess and have poor adherence to real clinical picture of the two diseases.

Answer: The authors would like to thank the reviewers for their valuable input. We have rephrased the sentences as per your requirements.

Clinically, IBD features disclose a variety of ranges based on age groups and gender basis, such as more than 60% of women suffering from CD are reported with rectal bleeding. Conversely, 62% of UC cases in males exhibit reduced rectal bleeding and abdominal pain events [1]. The results of prevalence data show that nearly 1 million people in the USA were reported with CD. Interestingly, more than 80% of reported cases of CD reveal that the patients' distal part of the small intestine is affected [2]. Nevertheless, patients suffering from UC appeared on the board with inflammatory lesions, especially in the distal part of the colon [3].

-        * “Thus, NAFLD, known as MAFLD”

NALFD is the old term. Now the term is MAFLD

Answer: The authors would like to thank the reviewers for the comment. The term NAFLD is old, but the article we cited used the word NAFLD, so we didn’t change the term to acknowledge the efforts of that article. NAFLD is still a prevailing term and is being accepted by the majority of scientists, while MAFLD is still emerging with a bunch of conflicts. Conversely, we briefly described NAFLD as MAFLD between 70-73 line numbers.

-       - Increase the size of the characters of Figure 1, 4

Answer:  The authors thank the reviewers for their comments. We have increased the size of the characters in figure 1 and 4

-        -The treatments of IBD in Table 2 are randomly ordered and randomly chosen (incomplete)

Answer:  The authors thank the reviewers for the comment. The items in Table 2 were arranged according to the subclass of the clinical availability (Mab drugs, small molecules, and new drugs). The authors would like to keep the table as it is now.

-        - A clinician co-author expert in IBD could be useful in improving the clinical part of the review

Answer: The authors thank the reviewer for their recommendation. However, the theme of the articles primarily relates to the molecular mechanisms not oriented around clinical evaluation. As clinicians have a good grip on clinical aspects as compared to molecular aspects, we briefly discussed clinical aspects and didn’t add clinicians to the panel

-        - “The standard therapy for IBD patients typically involves the use of aminosalicylates” Are you talking about ulcerative colitis or Crohn’s disease?

Answers: The authors would like to thank the reviewers for their valuable input. The theme of this review is majorly focusing on the interrelationships of IBD with liver disease, irrespective of Crohn’s disease or Ulcerative colitis. Because of that, we discussed both CD and UC briefly under the umbrella of IBD.

-        - “The result of the tocilizumab trial on 36 patients 463 with CD has been reported with clinical significance [180]” Tocilizumab failed in IBD

Answers: The authors would like to thank the reviewers for their efforts. To date, I haven’t found an article stating that tocilizumab has failed in IBD. I would appreciate it enormously if you could provide any scientific article references regarding the above-mentioned statement by you, and we will be happy to cite those articles.

-        - “However, S1P receptor 495 modulators have shown promise in preclinical studies [192] and are currently being 496 evaluated in clinical trials for inflammatory disorders.” Not updated data

Answers: The authors would like to thank the reviewer's efforts. We have updated the information for your consideration.

Recently, it was reported that ozanimod has been in phase II for CD and phase III for UC treatment. Etrasimod is currently in phase II trials for UC, while amiselimod has completed phase II trials for CD [4].  

References

  1. Charpentier, C.; Salleron, J.; Savoye, G.; Fumery, M.; Merle, V.; Laberenne, J.-E.; Vasseur, F.; Dupas, J.-L.; Cortot, A.; et al. Natural history of elderly-onset inflammatory bowel disease: a population-based cohort study. Gut 2014, 63, (3), 423-432. http://dx.doi.org/10.1136/gutjnl-2012-303864
  2. Kappelman, M. D.; Rifas–Shiman, S. L.; Kleinman, K.; Ollendorf, D.; Bousvaros, A.; Grand, R. J.; Finkelstein, J. A. The prevalence and geographic distribution of Crohn’s disease and ulcerative colitis in the United States. Clin Gastroenterol Hepatol 2007, 5, (12), 1424-1429. https://doi.org/10.1016/j.cgh.2007.07.012
  3. Magro, F.; Langner, C.; Driessen, A.; Ensari, A.; Geboes, K.; Mantzaris, G.; Villanacci, V.; Becheanu, G.; Nunes, P. B.; et al. European consensus on the histopathology of inflammatory bowel disease. J Crohns Colitis 2013, 7, (10), 827-851. https://doi.org/10.1016/j.crohns.2013.06.001
  4. Danese, S.; Furfaro, F.; Vetrano, S. Targeting S1P in inflammatory bowel disease: new avenues for modulating intestinal leukocyte migration. J Crohns Colitis 2018, 12, (suppl_2), S678-S686. https://doi.org/10.1093/ecco-jcc/jjx107

Dear Editor,

On behalf of the authors, I would like to express sincere thanks for your decision on our manuscript considered in the special issue of “ Metabolic Diseases and Complications in Association with Organ Cross-Talk.” We put more efforts into revising the MS according to the helpful comments of the referee(s). I appreciate your kind final consideration for disseminating our review paper in the International Journal of Molecular Sciences

Sincerely,

Professor Sang Geon Kim, Ph.D.

College of Pharmacy

Dongguk University-Seoul

Reviewer 2 Report

Comments and Suggestions for Authors

Kwon and colleagues present the manuscript for a review on the effects of dietary lipids on inflammatory bowel disease. The topic of the review is pertinent and overall, the authors have presented a well-written manuscript. Nevertheless, I found some issues with this review that should be addressed by the authors before publication.

Major issues

·         The title of the review suggests that it will deal in its majority with the impact and mechanisms of lipids on IBD. However, this topic only make up roughly 30% of the text. In contrast, much space is dedicated to molecular mechanisms and cytokines etc. in IBD (until page 9) and the end of the manuscript then deals with new therapeutic approaches for IBD. While the provided information is pertinent and interesting, this structure and emphasis of the review makes the title somewhat misleading. The authors should therefore either restructure their review to focus more on lipids and possibly other dietary components, OR rename their manuscript to have a more general title (preferred).

·         Table 1 is references in line 62, implying it has something to do with “Any pathological disruption due to chronic caloric intake results in metabolic syn-61 drome (i.e., diabetes, obesity, and fatty liver disease)” (sentence before). However, table 1 shows “studies on IBD and obesity risk” (which also seems misnamed, should it not be “Association of obesity and BD risk”?). This information however does not fit with the text here. Please revise where this table should be references correctly.

·         In Figure 1, RORgt comes out of nowhere as it is only mentioned much further down in the review. Please explain this in the text or in the figure legend directly. The connection here is also unclear.

·         In line 169, the authors write that the major role of IL-6 is to provoke “anti-inflammatory impacts”. While in my opinion, there is no such thing as an “anti-inflammatory impact”, IL-6 also very much acts as a PRO-inflammatory cytokine, so I am a bit confused what the authors want to say here. Please revise.

·         As mentioned above, only Lines 347 – 451 actually deal with the topic implied in the title of the manuscript, whereas the following section is 4 is actually unrelated.

·         The contradictory finding that many IBD patients are obese and that the obesity might actually be an effect of IBD (rather than obesity being the cause for IBD) should be discussed in much more detail as this is part of the core topic of the review.

Minor issues

While the manuscript is in general well written, there are some typos, duplications etc. This is not an exhaustive list and the manuscript should be carefully language edited before publication.

·         Line 54: The sentence starting with “The liver abundantly…” is almost an exact duplication of the sentence in line 48. Please revise.

·         Line 61: I suppose the authors mean chronic “excessive” caloric intake?

·         Line 71: Please explain “NAFLD, known as MAFLD”. This is not used anywhere else and is rather confusing.

·         Line 76: The sentence starting with “The recent era of…” is not understandable. Please revise.

·         The abbreviation IEC is not defined. Sometimes the full version is written out and sometimes the abbreviation is used. Please define at the first instance and then stick to the abbreviation.

·         Line 111: What is “candid disclosure”?

·         Line 125 and others. Please stick to using TNF-a instead of just writing TNF

·         Line 108: “target genes’ transcription” (apostrophe missing)

·         Line 307 and 309: delete “the” before “in vivo” and “alternate”

·         Line 336: effector

·         Table 2:” inhibits”

·         Figure 2 seems to be resized non-proportionally. Please revise.

·         Figure 4: Maybe add a crossed out MGAT2 on top of the disrupted epithelial cells. Otherwise, it takes quite a while to understand the meaning of the figure.

·         Figure 5: “flowchart” or “scheme” instead of “schematic flow”

·         Line 234: Is the reference to Figure 1 correct here? Figure contains nothing about HDAC3 and C4 T cells. Is there a figure missing?

·         “Perspectives” seems the wrong choice for the title of the last section, as the provided information is more of a summary rather than a look into the future. I recommend calling this section simply “Conclusions”.

Comments on the Quality of English Language

Good review with some editing required. The title should be changed as it is misleading with regards to the contents of the review.

Author Response

Reviewer 2

Comments and Suggestions for Authors:  Kwon and colleagues present the manuscript for a review on the effects of dietary lipids on inflammatory bowel disease. The topic of the review is pertinent and overall, the authors have presented a well-written manuscript. Nevertheless, I found some issues with this review that should be addressed by the authors before publication.

Major issues

  • The title of the review suggests that it will deal in its majority with the impact and mechanisms of lipids on IBD. However, this topic only make up roughly 30% of the text. In contrast, much space is dedicated to molecular mechanisms and cytokines etc. in IBD (until page 9) and the end of the manuscript then deals with new therapeutic approaches for IBD. While the provided information is pertinent and interesting, this structure and emphasis of the review makes the title somewhat misleading. The authors should therefore either restructure their review to focus more on lipids and possibly other dietary components, OR rename their manuscript to have a more general title (preferred).

Answer: The authors would like to thank the reviewers for their intellectual input. Precisely, the immunological response is activated via infections and also with dietary lipids. We don’t want to insist on the already well-established data to avoid repetition. We focused on how activated immunological responses affect IBD and liver diseases and tried to develop a relationship between liver pathology and IBD.

We described the introductory statements on lipid accumulation in the liver as follows to cover lipid effects on IBD: “In recent days, NAFLD progression belongs to the chronic caloric intake. Based on hospitalization diagnosis, IBD patients have high body mass index because of high-fat diet (HFD) consumption, which exhibits deleterious effects [1].”

  • Table 1 is references in line 62, implying it has something to do with “Any pathological disruption due to chronic caloric intake results in metabolic syn-61 drome (i.e., diabetes, obesity, and fatty liver disease)” (sentence before). However, table 1 shows “studies on IBD and obesity risk” (which also seems misnamed, should it not be “Association of obesity and BD risk”?). This information however does not fit with the text here. Please revise where this table should be references correctly.

Answer: The authors would like to thank the reviewer's efforts. We renamed the title of the table.

  • In Figure 1, RORgt comes out of nowhere as it is only mentioned much further down in the review. Please explain this in the text or in the figure legend directly. The connection here is also unclear.

Answer: The authors would like to thank the reviewers. We added the portion in the text before figure 1. Retinoic acid–related orphan receptor-γt (RORγt) is a transcription factor that is involved in the development of TH17 [2].

  • In line 169, the authors write that the major role of IL-6 is to provoke “anti-inflammatory impacts”. While in my opinion, there is no such thing as an “anti-inflammatory impact”, IL-6 also very much acts as a PRO-inflammatory cytokine, so I am a bit confused what the authors want to say here. Please revise.

Answer: The authors would like to thank the reviewers. The given information is correct you can validate it from cited references in the text.

IL‑6 is generated by several cells present inside the tumor, such as tumor-infiltrating cells and stromal cells. IL-6 in normal blood concentration (1.6 pg/ml) facilitates a mild immune response against the defense of incessant pathogens [3]. Studies have proven that IL-6 association with the vagus nerve may have effects on the smooth muscle cells or secretory cells, which results in intestinal motility and secretion [3, 4]. The classic pathway of IL-6 signaling includes the binding of IL-6 with the membrane-bound receptor IL-6 receptor-a, also known as IL-6R. This binding results in the development of a heterohexameric complex comprising two IL-6, IL-6R, and the b subunit of IL-6 receptor (gp130) [5, 6]. This complex then leads to the stimulation of the JAK/STAT3 pathway, consequently integrating STAT3 target genes (Fig. 2). Interestingly, the complex also triggers the PI3K/AKT/mTOR and RAS/RAF/MEK/ERK pathways [7]. The major role of the classical pathway is to provoke anti-inflammatory impacts during the acute-phase response [8].

  • As mentioned above, only Lines 347 – 451 actually deal with the topic implied in the title of the manuscript, whereas the following section is 4 is actually unrelated.

Answer: The authors would like to thank the reviewers. The purpose of section 4 is to emphasize the immunological treatment and its beneficial effects, which could also be helpful in liver disease.

  • The contradictory finding that many IBD patients are obese and that the obesity might actually be an effect of IBD (rather than obesity being the cause for IBD) should be discussed in much more detail as this is part of the core topic of the review. 

Answer: The author would like to thank the reviewer. We briefly added the interrelationship of obesity with the modulation of the microbiome.

Obesity is usually associated with a variety of metabolic syndromes, such as type 2 diabetes [9], ischemic vascular disease [10], elevated serum lipids levels, and NAFLD [11]. Studies report that the experimental animal models, i.e., leptin-deficient ob/ob mice, can alter the microbiome [12] to develop insulin resistance and diabetes via the regulation of several molecular cascades, such as altered fatty acid metabolism in the liver and modulation of the glucagon-like peptide [13].

Minor issues

While the manuscript is in general well written, there are some typos, duplications etc. This is not an exhaustive list and the manuscript should be carefully language edited before publication.

Answer: The authors would like to thank the reviewer's comment. We keenly checked the English mistakes.

  • Line 54: The sentence starting with “The liver abundantly…” is almost an exact duplication of the sentence in line 48. Please revise.

Answer: The authors would like to thank the reviewers for their comments. We corrected the line.

  • Line 61: I suppose the authors mean chronic “excessive” caloric intake?

Answer: The authors would like to thank the reviewers for their comments. Chronic and excessive generally are the same terms however, technically chronic caloric intake means the persistent excessive use of caloric intake.

  • Line 71: Please explain “NAFLD, known as MAFLD”. This is not used anywhere else and is rather confusing.

Answer: The authors would like to thank the reviewers for their comments. The term NAFLD is old, but the article we cited used the word NAFLD, so we didn’t change the term to acknowledge the efforts of that article. NAFLD is still a prevailing term and is being accepted by most scientists, while MAFLD is still emerging with many conflicts. Conversely, we briefly described NAFLD as MAFLD between 70-73 line numbers.

  • Line 76: The sentence starting with “The recent era of…” is not understandable. Please revise.

Answer: The authors would like to thank the reviewers. We have revised the line for your consideration.

In recent days, NAFLD progression belongs to the chronic caloric intake. Based on hospitalization diagnosis, IBD patients have high body mass index because of high-fat diet (HFD) consumption, which exhibits deleterious effects [1].

  • The abbreviation IEC is not defined. Sometimes the full version is written out and sometimes the abbreviation is used. Please define at the first instance and then stick to the abbreviation.

Answer: The authors would like to thank the reviewers. When we are writing an article, we don’t always use the full form or abbreviation; we intermix these words as per condition.

  • Line 111: What is “candid disclosure”?

Answer: The author thanks the reviewers for their efforts to improve the quality of the manuscript. Candid disclosure is a term generally used to put emphasis on the harmful/negative effects

  • Line 125 and others. Please stick to using TNF-a instead of just writing TNF

Answer: The authors would like to thank the reviewers. TNF is the major superfamily, which includes TNF-a and TNF-related apoptosis ligands. Because of that, we used two different terms

  • Line 108: “target genes’ transcription” (apostrophe missing)

Answer: The authors would like to thank the reviewer. That word was edited for your consideration.

  • Line 307 and 309: delete “the” before “in vivo” and “alternate”

Answer: The authors would like to thank the reviewers. We rephrased the line for your consideration.

The results of the in vivo experiments show that the intestinal epithelium is sustained after Paneth cell depletion, which supports the alternate cascade activation

  • Line 336: effector

Answer: The authors would like to thank the reviewer. That word was edited for your consideration.

  • Table 2:” inhibits”

Answer: The authors would like to thank the reviewers. With your suggested changes, the sentence will be grammatically incorrect. But we modified the sentence: Inhibition of the enzymes responsible for nucleotide synthesis

  • Figure 2 seems to be resized non-proportionally. Please revise.

Answer:  The authors would like to thank the reviewers. Figure 2 was resized and adjusted by the journal so we can not apply that change.

  • Figure 4: Maybe add a crossed out MGAT2 on top of the disrupted epithelial cells. Otherwise, it takes quite a while to understand the meaning of the figure.

Answer: The authors would like to thank the reviewers. Figure 4 is placed appropriately in the context of the data.

  • Figure 5: “flowchart” or “scheme” instead of “schematic flow”

Answer: The authors would like to thank the reviewers. Figure 5 is the summary of the article to ensure the concept, it is necessary to add the schematic flow to better understand the concept.

  • Line 234: Is the reference to Figure 1 correct here? Figure contains nothing about HDAC3 and C4 T cells. Is there a figure missing?

Answer: The authors would like to thank the reviewers. That was a typographical mistake which we corrected by removing it.

  • “Perspectives” seems the wrong choice for the title of the last section, as the provided information is more of a summary rather than a look into the future. I recommend calling this section simply “Conclusions”.

Answer: The authors would like to thank the reviewers. We renamed perspective as conclusion.

Comments on the Quality of English Language

Good review with some editing required. The title should be changed as it is misleading with regards to the contents of the review.

Answer: The authors would like to thank the reviewer. However, the manuscript already explained the portion regarding the lipids relation by explaining NAFLD, obesity, adding a portion of lipids and inflammation, and also dietary lipids and IBD progression. The authors would like to keep the title as it is.

References

  1. Blain, A.; Cattan, S.; Beaugerie, L.; Carbonnel, F.; Gendre, J.; Cosnes, J. Crohn's disease clinical course and severity in obese patients. Clin Nutr 2002, 21, (1), 51-57. https://doi.org/10.1054/clnu.2001.0503
  2. Rizzo, A.; Di Giovangiulio, M.; Stolfi, C.; Franzè, E.; Fehling, H.-J.; Carsetti, R.; Giorda, E.; Colantoni, A.; Ortenzi, A.; et al. RORγt-Expressing Tregs Drive the Growth of Colitis-Associated Colorectal Cancer by Controlling IL6 in Dendritic CellsRORγt+ Tregs Drive CAC by Controlling IL6 in DCs. Cancer Immunol Res 2018, 6, (9), 1082-1092. https://doi.org/10.1158/2326-6066.CIR-17-0698
  3. Wennerås, C.; Hagberg, L.; Andersson, R.; Hynsjö, L.; Lindahl, A.; Okroj, M.; Blom, A. M.; Johansson, P.; Andreasson, B.; et al. Distinct inflammatory mediator patterns characterize infectious and sterile systemic inflammation in febrile neutropenic hematology patients. PloS one 2014, 9, (3), e92319. https://doi.org/10.1371/journal.pone.0092319
  4. Comini, L.; Pasini, E.; Bachetti, T.; Dreano, M.; Garotta, G.; Ferrari, R. Acute haemodynamic effects of IL-6 treatment in vivo: involvement of vagus nerve in NO-mediated negative inotropism. Cytokine 2005, 30, (5), 236-242. https://doi.org/10.1016/j.cyto.2005.01.009
  5. Hunter, C. A.; Jones, S. A. IL-6 as a keystone cytokine in health and disease. Nat Immunol 2015, 16, (5), 448-457. https://doi.org/10.1038/ni.3153
  6. Ohshima, S.; Saeki, Y.; Mima, T.; Sasai, M.; Nishioka, K.; Nomura, S.; Kopf, M.; Katada, Y.; Tanaka, T.; et al. Interleukin 6 plays a key role in the development of antigen-induced arthritis. Proc Natl Acad Sci USA 1998, 95, (14), 8222-8226. https://doi.org/10.1073/pnas.95.14.8222
  7. Yeh, Y.-H.; Hsiao, H.-F.; Yeh, Y.-C.; Chen, T.-W.; Li, T.-K. Inflammatory interferon activates HIF-1α-mediated epithelial-to-mesenchymal transition via PI3K/AKT/mTOR pathway. J Exp Clin Cancer Res 2018, 37, 1-15. https://doi.org/10.1186/s13046-018-0730-6
  8. Baran, P.; Hansen, S.; Waetzig, G. H.; Akbarzadeh, M.; Lamertz, L.; Huber, H. J.; Ahmadian, M. R.; Moll, J. M.; Scheller, J. The balance of interleukin (IL)-6, IL-6· soluble IL-6 receptor (sIL-6R), and IL-6· sIL-6R· sgp130 complexes allows simultaneous classic and trans-signaling. J Biol Chem 2018, 293, (18), 6762-6775. https://doi.org/10.1074/jbc.RA117.001163
  9. Quan, W.; Jung, H. S.; Lee, M.-S. Role of autophagy in the progression from obesity to diabetes and in the control of energy balance. Arch Pharm Res 2013, 36, 223-229. https://doi.org/10.1007/s12272-013-0024-7
  10. Koliaki, C.; Liatis, S.; Kokkinos, A. Obesity and cardiovascular disease: revisiting an old relationship. Metabolism 2019, 92, 98-107. https://doi.org/10.1016/j.metabol.2018.10.011
  11. Khan, M. S.; Lee, C.; Kim, S. G. Non-alcoholic fatty liver disease and liver secretome. Arch Pharm Res 2022, 1-26. https://doi.org/10.1007/s12272-022-01419-w
  12. Ley, R. E.; Bäckhed, F.; Turnbaugh, P.; Lozupone, C. A.; Knight, R. D.; Gordon, J. I. Obesity alters gut microbial ecology. Proc Natl Acad Sci USA 2005, 102, (31), 11070-11075. https://doi.org/10.1073/pnas.0504978102
  13. Musso, G.; Gambino, R.; Cassader, M. Obesity, diabetes, and gut microbiota: the hygiene hypothesis expanded? Diabetes Care 2010, 33, (10), 2277-2284. https://doi.org/10.2337/dc10-0556

Dear Editor,

On behalf of the authors, I would like to express sincere thanks for your decision on our manuscript considered in the special issue of “ Metabolic Diseases and Complications in Association with Organ Cross-Talk.” We put more efforts into revising the MS according to the helpful comments of the referee(s). I appreciate your kind final consideration for disseminating our review paper in the International Journal of Molecular Sciences

Sincerely,

Professor Sang Geon Kim, Ph.D.

College of Pharmacy

Dongguk University-Seoul

Round 2

Reviewer 1 Report

Comments and Suggestions for Authors

I have already rejected this article.

Comments on the Quality of English Language

I have already rejected this article.